# Dynamic Thinking-Token Selection for Efficient Reasoning in Large Reasoning Models

**Zhenyuan Guo** [1]  **Tong Chen** [1]  **Wenlong Meng** [1]  **Chen Gong** [2]  **Xin Yu** [3]  **Chengkun Wei** [1][†]  **Wenzhi Chen** [1]

## Abstract

Large Reasoning Models (LRMs) excel at solving complex problems by explicitly generating a reasoning trace before deriving the final answer. However, these extended generations incur substantial memory footprint and computational overhead, bottlenecking LRMs' efficiency. This work uses attention maps to analyze the influence of reasoning traces and uncover an interesting phenomenon: *only some decision-critical tokens in a reasoning trace steer the model toward the final answer, while the remaining tokens contribute negligibly.* Building on this observation, we propose **Dyn**amic **T**hinking-Token **S**election (DYNTS). This method identifies decision-critical tokens and retains only their associated Key-Value (KV) cache states during inference, evicting the remaining redundant entries to optimize efficiency. Across six benchmarks, DYNTS surpasses the state-of-the-art KV cache compression methods, improving Pass@1 by 2.6% under the same budget. Compared to vanilla Transformers, it reduces inference latency by $1.84$–$2.62\times$ and peak KV-cache memory footprint by $3.32$–$5.73\times$ without compromising LRMs' reasoning performance. The code is available at the link.[1]

## 1. Introduction

Recent advancements in Large Reasoning Models (LRMs) (Chen et al., 2025) have significantly strengthened the reasoning capabilities of Large Language Models (LLMs). Representative models such as DeepSeek-R1 (Guo et al., 2025), Gemini-3-Pro (DeepMind, 2025), and ChatGPT-5.2 (OpenAI, 2025) support deep thinking mode to strengthen reasoning capability in the challenging mathematics, programming, and science tasks (Zhang et al., 2025b). These models spend a substantial number of intermediate thinking tokens on reflection, reasoning, and verification to derive the correct response during inference (Feng et al., 2025). However, the thinking process necessitates the immense KV cache memory footprint and attention-related computational cost, posing a critical deployment challenge in resource-constrained environments.

KV cache compression techniques aim to optimize the cache state by periodically evicting non-essential tokens (Shi et al., 2024; WEI et al., 2025; Liu et al., 2025b; Qin et al., 2025), typically guided by predefined token retention rules (Chen et al., 2024; Xiao et al., 2024; Devoto et al., 2024) or attention-based importance metrics (Zhang et al., 2023; Li et al., 2024; Choi et al., 2025). Nevertheless, incorporating them into the inference process of LRMs faces two key limitations: (1) Methods designed for long-context prefilling are ill-suited to the short-prefill and long-decoding scenarios of LRMs; (2) Methods tailored for long-decoding struggle to match the reasoning performance of the Full KV baseline (SOTA 60.9% vs. Full KV 63.6%, Fig. 1 Left). Specifically, in LRM inference, the model conducts an extensive reasoning process and then summarizes the reasoning content to derive the final answer (Minegishi et al., 2025). This implies that the correctness of the final answer relies on the thinking tokens within the preceding reasoning (Bogdan et al., 2025). However, existing compression methods cannot identify the tokens that are essential to the future answer. This leads to a significant misalignment between the retained tokens and the critical thinking tokens, resulting in degradation in the model's reasoning performance.

To address this issue, we analyze the LRM's generated content and study which tokens are most important for the model to steer the final answer. Some works point out attention weights capturing inter-token dependencies (Vaswani et al., 2017; Wiegreffe & Pinter, 2019; Bogdan et al., 2025), which can serve as a metric to assess the importance of tokens. Consequently, we decompose the generated content into a reasoning trace and a final answer, and then calculate the importance score of each thinking token in the trajec-

[1]Department of Computer Science and Technology, Zhejiang University, Hangzhou, China [2]University of Virginia, Charlottesville, Virginia, USA [3]Institute of Information Processing and Intelligent Computing, Ningbo Tech University, Ningbo, China. Correspondence to: Chengkun Wei <weichengkun@zju.edu.cn>.

*Proceedings of the $43^{rd}$ International Conference on Machine Learning*, Seoul, South Korea. PMLR 306, 2026. Copyright 2026 by the author(s).

[1]https://github.com/Robin930/DynTS

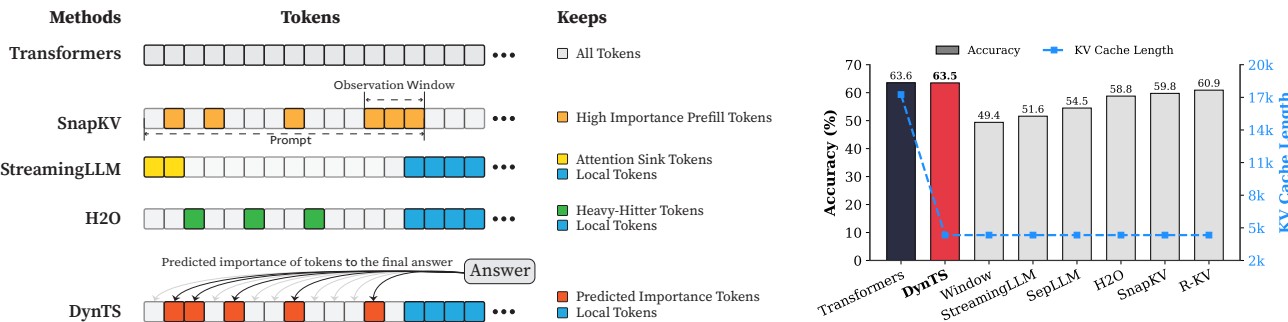

*Figure 1.* (Left) Comparison of token selection strategies across different KV cache eviction methods. In each row, colored blocks denote the retained high-importance tokens, while grey blocks represent the evicted tokens during LRM inference. (Right) The average reasoning performance and KV cache memory footprint on DeepSeek-R1-Distall-Llama-8B and DeepSeek-R1-Distall-Qwen-7B across six reasoning benchmarks.

tory by aggregating the attention weights from the answer to thinking tokens. We find that only a small subset of thinking tokens ($\sim 20\%$ tokens in the reasoning trace, see Section §3.1) have significant scores, which may be critical for the final answer. To validate these hypotheses, we retain these tokens and prompt the model to directly generate the final answer. Experimental results show that the model maintains close accuracy compared to using the whole KV cache. This reveals a Pareto principle[2] in LRMs: *only a small subset of decision-critical thinking tokens with high importance scores drives the model toward the final answer, while the remaining tokens contribute negligibly.*

Based on the above insight, we introduce **DYNTS** (**Dyn**amic **T**hinking-**T**oken **S**election), a novel method for dynamically predicting and selecting decision-critical thinking tokens on-the-fly during decoding, as shown in Fig. 1 (Left). The key innovation of DYNTS is the integration of a trainable, lightweight *Importance Predictor* at the final layer of LRMs, enabling the model to dynamically predict the importance of each thinking token to the final answer. By utilizing importance scores derived from sampled reasoning traces as supervision signals, the predictor learns to distinguish between critical tokens and redundant tokens. During inference, DYNTS manages memory through a dual-window mechanism: generated tokens flow from a Local Window (which captures recent context) into a Selection Window (which stores long-term history). Once the KV cache reaches the budget, the system retains the KV cache of tokens with higher predicted importance scores in the Select Window and all tokens in the Local Window (Zhang et al., 2023; Chen et al., 2024). By evicting redundant KV cache entries, DYNTS effectively reduces both system memory pressure and computational overhead. We also theoretically analyze the computational overhead introduced by the importance

predictor and the savings from cache eviction, and derive a Break-Even Condition for net computational gain.

Then, we train the Importance Predictor based on the MATH (Hendrycks et al., 2021) train set, and evaluate DYNTS on the other six reasoning benchmarks. The reasoning performance and KV cache length compare with the SOTA KV cache compression method, as reported in Fig. 1 (Right). Our method reduces the KV cache memory footprint by up to $3.32$–$5.73\times$ without compromising reasoning performance compared to the full-cache transformer baseline. Within the same budget, our method achieves a $2.6\%$ improvement in accuracy over the SOTA KV cache compression approach.

## 2. Preliminaries

**Large Reasoning Model (LRM).** Unlike standard LLMs that directly generate answers, LRMs incorporate an intermediate reasoning process prior to producing the final answer (Chen et al., 2025; Zhang et al., 2025a; Sui et al., 2025). Given a user prompt $\mathbf{x} = (x_1, \dots, x_M)$, the model generated content represents as $\mathbf{y}$, which can be decomposed into a reasoning trace $\mathbf{t}$ and a final answer $\mathbf{a}$. The trajectory is delimited by a start tag <think> and an end tag </think>. Formally, the model output is defined as:

$$\mathbf{y} = [\texttt{<think>}, \mathbf{t}, \texttt{</think>}, \mathbf{a}], \tag{1}$$

where the trajectory $\mathbf{t} = (t_1, \dots, t_L)$ composed of $L$ thinking tokens, and $\mathbf{a} = (a_1, \dots, a_K)$ represents the answer composed of $K$ tokens. During autoregressive generation, the model conducts a reasoning phase that produces thinking tokens $t_i$, followed by an answer phase that generates the answer token $a_i$. This process is formally defined as:

$$P(\mathbf{y}|\mathbf{x}) = \underbrace{\prod_{i=1}^{L} P(t_i|\mathbf{x}, \mathbf{t}_{<i})}_{\text{Reasoning Phase}} \cdot \underbrace{\prod_{j=1}^{K} P(a_j|\mathbf{x}, \mathbf{t}, \mathbf{a}_{<j})}_{\text{Answer Phase}} \tag{2}$$

---

[2]The Pareto principle, also known as the 80/20 rule, posits that 20% of critical factors drive 80% of the outcomes. In this paper, it implies that a small fraction of pivotal thinking tokens dictates the correctness of the model's final response.

Since the length of the reasoning trace significantly exceeds that of the final answer ($L \gg K$) (Xu et al., 2025), we focus on selecting critical thinking tokens in the reasoning trace to reduce memory and computational overhead.

**Attention Mechanism.** Attention Mechanism is a core component of Transformer-based LRMs, such as Multi-Head Attention (Vaswani et al., 2017), Grouped-Query Attention (Ainslie et al., 2023), and their variants. To highlight the memory challenges in LRMs, we formulate the attention computation at the token level. Consider the decode step $t$. Let $\mathbf{h}_t \in \mathbb{R}^d$ be the input hidden state of the current token. The model projects $\mathbf{h}_t$ into query, key, and value vectors:

$$\mathbf{q}_t = \mathbf{W}_Q \mathbf{h}_t, \quad \mathbf{k}_t = \mathbf{W}_K \mathbf{h}_t, \quad \mathbf{v}_t = \mathbf{W}_V \mathbf{h}_t, \quad (3)$$

where $\mathbf{W}_Q, \mathbf{W}_K, \mathbf{W}_V$ are learnable projection matrices. The query $\mathbf{q}_t$ attends to the keys of all preceding positions $j \in \{1, \ldots, t\}$. The attention weight $a_{t,j}$ between the current token $t$ and a past token $j$ is:

$$\alpha_{t,j} = \frac{\exp(e_{t,j})}{\sum_{i=1}^{t} \exp(e_{t,i})}, \qquad e_{t,j} = \frac{\mathbf{q}_t^\top \mathbf{k}_j}{\sqrt{d_k}}. \quad (4)$$

These scores represent the relevance of the current step to the $j$-th token. Finally, the output of the attention head $\mathbf{o}_t$ is the weighted sum of all historical value vectors:

$$\mathbf{o}_t = \sum_{j=1}^{t} \alpha_{t,j} \mathbf{v}_j. \quad (5)$$

As Equation (5) implies, calculating $\mathbf{o}_t$ requires access to the entire sequence of past keys and values $\{\mathbf{k}_j, \mathbf{v}_j\}_{j=1}^{t-1}$. In standard implementation, these vectors are stored in the KV cache to avoid redundant computation (Vaswani et al., 2017; Pope et al., 2023). In the LRMs' inference, the reasoning trace is exceptionally long, imposing significant memory bottlenecks and increasing computational overhead.

## 3. Observations and Insight

This section presents the observed sparsity of thinking tokens and the Pareto Principle in LRMs, serving as the basis for DYNTS. Detailed experimental settings and additional results are provided in Appendix §B.

### 3.1. Sparsity for Thinking Tokens

Previous works (Bogdan et al., 2025; Zhang et al., 2023; Singh et al., 2024) have shown that attention weights (Eq. 4) serve as a reliable proxy for token importance. Building on this insight, we calculate an importance score for each question and thinking token by accumulating the attention they receive from all answer tokens. Formally, the importance

scores are defined as:

$$I_{x_j} = \sum_{i=1}^{K} \alpha_{a_i, x_j}, \qquad I_{t_j} = \sum_{i=1}^{K} \alpha_{a_i, t_j}, \qquad (6)$$

where $I_{x_j}$ and $I_{t_j}$ denote the importance scores of the $j$-th question token $x_j$ and thinking token $t_j$. Here, $\alpha_{a_i, x_j}$ and $\alpha_{a_i, t_j}$ represent the attention weights from the $i$-th answer token $a_i$ to the corresponding question or thinking token, and $K$ is the total number of answer tokens. We perform full autoregressive inference on LRMs to extract attention weights and compute token-level importance scores for both question and thinking tokens.

**Observation.** As illustrated in Fig. 2, the question tokens (left panel) exhibit consistently significant and dense importance scores. In contrast, the thinking tokens (right panel) display a highly sparse distribution. Despite the extensive reasoning trace (exceeding 12k tokens), only 21.1% of thinking tokens exceed the mean importance score. This indicates that the vast majority of reasoning steps exert only a marginal influence on the final answer.

**Analysis.** Follow attention-based methods (Cai et al., 2025; Li et al., 2024; Cai et al., 2024), tokens with higher importance scores intuitively correspond to decision-critical reasoning steps, which are critical for the model to generate the final answer. The low-importance tokens serve as syntactic scaffolding or intermediate states that become redundant after reasoning progresses (We report the ratio of *Content Words*, see Appendix B.2). Consequently, we hypothesize that the model maintains close reasoning performance to that of the full token sequence, even when it selectively retains only these critical thinking tokens.

### 3.2. Pareto Principle in LRMs

To validate the aforementioned hypothesis, we retain all question tokens while preserving only the top-$p\%$ of thinking tokens ranked by importance score, and prompt the model to directly generate the final answer.

**Observation.** As illustrated in Fig. 3 (Left), the importance-based top-$p\%$ selection strategy substantially outperforms both random- and bottom-selection baselines. Notably, the model recovers nearly its full performance (grey dashed line) when retaining only $\sim 30\%$ thinking tokens with top importance scores. Fig. 3 (Right) further confirms this trend across six diverse datasets, where the performance polygon under the top-30% retention strategy almost completely overlaps with the full thinking tokens.

**Insights.** These empirical results illustrate and reveal the Pareto Principle in LRM reasoning: *Only a small subset of thinking tokens ( 30%) with high importance scores serve as "pivotal nodes," which are critical for the model to output a final answer, while the remaining tokens contribute negligi-*

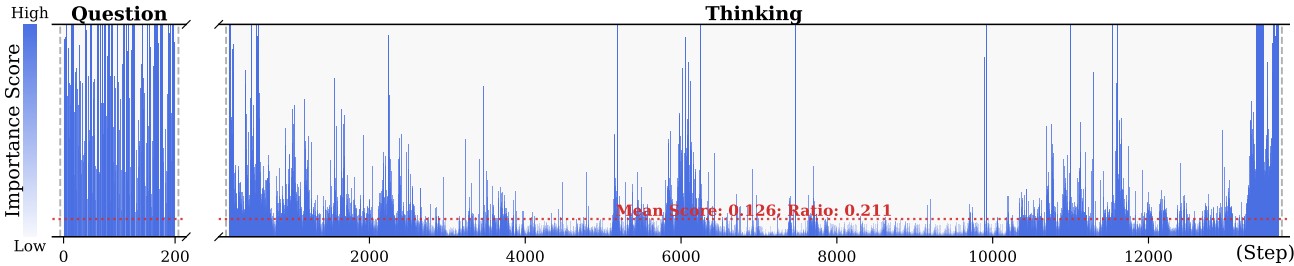

*Figure 2.* Importance scores of question tokens and thinking tokens in a reasoning trace, computed based on attention contributions to the answer. Darker colors indicate higher importance. The red dashed line shows the mean importance score, and the annotated ratio indicates the fraction of tokens with importance above the mean.

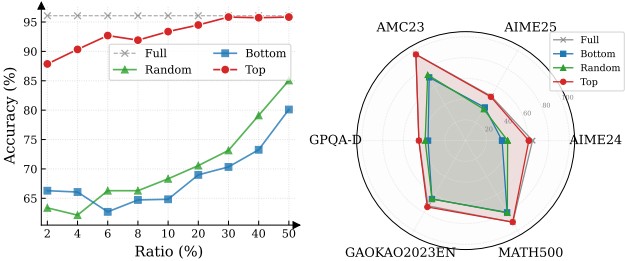

*Figure 3.* (Left) Reasoning performance trends as a function of thinking token retention ratio, where the $x$-axis indicates the retention percentage and the $y$-axis is the accuracy. (Right) Accuracy across all datasets when retaining 30% of the thinking tokens.

*bly to the outcome.* This finding provides strong empirical support for LRMs' KV cache compression, indicating that it is possible to reduce memory footprint and computational overhead without sacrificing performance.

## 4. Dynamic Thinking-Token Selection

Building on the Pareto Principle in LRMs, critical thinking tokens can be identified via the importance score computed by Equation (6). However, this computation requires the attention weights from the answer to the thinking tokens, which are inaccessible until the model completes the entire decoding stage. To address this limitation, we introduce an *Importance Predictor* that dynamically estimates the importance score of each thinking token during inference time. Furthermore, we design a decoding-time *KV cache Selection Strategy* that retains critical thinking tokens and evicts redundant ones. We refer to this approach as **DYNTS** (**Dyn**amic **T**hinking **T**oken **S**election), and the overview is illustrated in Fig. 4.

### 4.1. Importance Predictor

**Integrate Importance Predictor in LRMs.** Transformer-based Large Language Models (LLMs) typically consist of stacked Transformer blocks followed by a language modeling head (Vaswani et al., 2017), where the output of the final block serves as a feature representation of the current token. Building on this architecture, we attach an additional lightweight MLP head to the final hidden state, named as *Importance Predictor* (Huang et al., 2024). It is used to predict the importance score of the current thinking token during model inference, capturing its contribution to the final answer. Formally, we define the modified LRM as a mapping function $\mathcal{M}$ that processes the input sequence $\mathbf{x}_{\leq t}$ to produce a dual-output tuple comprising the next token $x_{t+1}$ and the current importance score $s_{x_t}$:

$$\mathcal{M}(\mathbf{x}_{\leq t}) \rightarrow (x_{t+1}, s_{x_t}) \qquad (7)$$

**Predictor Training.** To obtain supervision signals for training, we prompt the LRMs based on the training dataset to generate complete sequences denoted as $\{x_{1...M}, t_{1...L}, a_{1...K}\}$, filtering out incorrect or incomplete reasoning. Here, $x$, $t$, and $a$ represent the question, thinking, and answer tokens, respectively. Based on the observation in Section §3, the thinking tokens significantly outnumber answer tokens ($L \gg K$), and question tokens remain essential. Therefore, DYNTS only focuses on predicting the importance of thinking tokens. By utilizing the attention weights from answer to thinking tokens, we derive the ground-truth importance score $I_{t_i}$ for each thinking token according to Equation (6). Finally, the Importance Predictor parameters can be optimized by minimizing the Mean Squared Error (MSE) loss (Wang & Bovik, 2009) as follows:

$$\mathcal{L}_{\text{MSE}} = \frac{1}{K} \sum_{i=1}^{K} (I_{t_i} - s_{t_i})^2. \qquad (8)$$

To preserve the LRMs' original performance, we freeze the backbone parameters and optimize the Importance Predictor exclusively. The trained model can predict the importance of thinking tokens to the answer. This paper focuses on mathematical reasoning tasks. We optimize the Importance Predictor only on the MATH training set and validated across six other datasets (See Section §6.1).

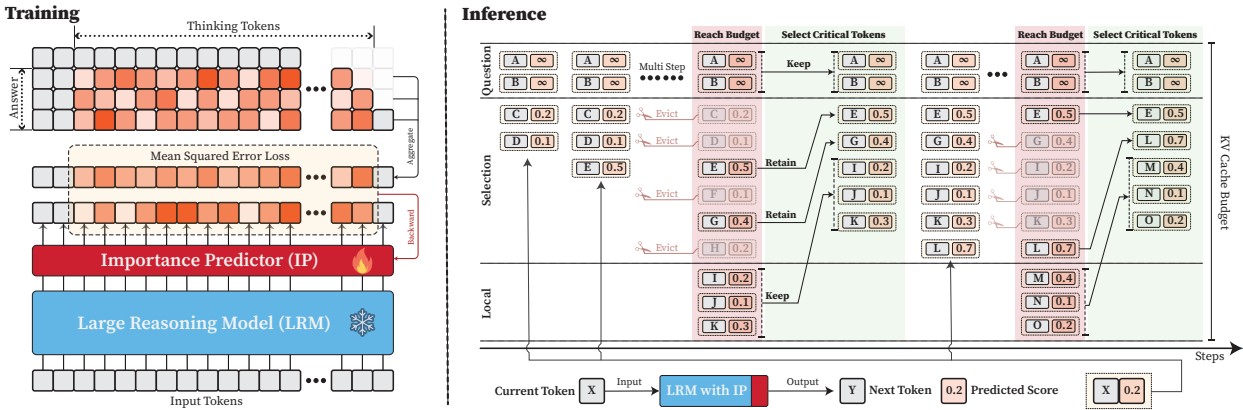

*Figure 4.* Overview of DYNTS. (Left) *Importance Predictor* Training. The upper heatmap visualizes attention weights, where orange intensity represents the importance of thinking tokens to the answer. The lower part shows a LRM integrated with an Importance Predictor (IP) to learn these importance scores. (Right) Inference with *KV Cache Selection*. The model outputs the next token and a predicted importance score of the current token. When the cache budget is reached, the selection strategy retains the KV cache of question tokens, local tokens, and top-k thinking tokens based on the predicted importance score.

### 4.2. KV Cache Selection

During LRMs' inference, we establish a maximum KV cache budget $B$, which is composed of a question window $W_q$, a selection window $W_s$, and a local window $W_l$, formulated as $B = W_q + W_s + W_l$. Specifically, the question window stores the KV caches of question tokens generated during the prefilling phase, i.e., the window size $W_q$ is equal to the number of question tokens $M$ ($W_q = M$). Since these tokens are critical for the final answer (see Section § 3), we assign an importance score of $+\infty$ to these tokens, ensuring their KV caches are immune to eviction throughout the inference process.

In the subsequent decoding phase, we maintain a sequential stream of tokens. Newly generated KV caches and their corresponding importance scores are sequentially appended to the selection window ($W_s$) and the local window ($W_l$). Once the total token count reaches the budget limit $B$, the critical token selection process is triggered, as illustrated in Fig. 4 (Right). Within the selection window, we retain the KV caches of the top-$k$ tokens with the highest scores and evict the remainder. Simultaneously, drawing inspiration from (Chen et al., 2024; Zhang et al., 2023; Zhao et al., 2024), we maintain the KV caches within the local window to ensure the overall coherence of the subsequently generated sequence. This inference process continues until decoding terminates.

## 5. Theoretical Overhead Analysis

In DYNTS, the KV cache selection strategy reduces computational overhead by constraining cache length, while the importance predictor introduces a slight overhead. In this section, we theoretically analyze the trade-off between these two components and derive the Break-even Condition required to achieve net computational gains.

**Notions.** Let $\mathcal{M}_{\text{base}}$ be the vanilla LRM with $L$ layers and hidden dimension $d$, and $\mathcal{M}_{\text{opt}}$ be the LRM with Importance Predictor (MLP: $d \rightarrow 2d \rightarrow d/2 \rightarrow 1$). We define the prefill length as $M$ and the current decoding step as $i \in \mathbb{Z}^+$. For vanilla decoding, the effective KV cache length grows linearly as $S_i^{\text{base}} = M + i$. While DYNTS evicts $K$ tokens by the KV Cache Selection when the effective KV cache length reaches the budget $B$. Resulting in the effective length $S_i^{\text{opt}} = M + i - n_i \cdot K$, where $n_i = \max\left(0, \left\lfloor \frac{(M+i)-B}{K} \right\rfloor + 1\right)$ denotes the count of cache eviction event at step $i$. By leveraging Floating-Point Operations (FLOPs) to quantify computational overhead, we establish the following theorem. The detailed proof is provided in Appendix A.

**Theorem 5.1** (Computational Gain). *Let $\Delta\mathcal{C}(i)$ be the reduction FLOPs achieved by DYNTS at decoding step $i$. The gain function is derived as the difference between the eviction event savings from KV Cache Selection and the introduced overhead of the predictor:*

$$\Delta\mathcal{C}(i) = \underbrace{n_i \cdot 4LdK}_{\textit{Eviction Saving}} - \underbrace{(6d^2 + d)}_{\textit{Predictor Overhead}} , \qquad (9)$$

Based on the formulation above, we derive a critical corollary regarding the net computational gain.

**Corollary 5.2** (Break-even Condition). *To achieve a net computational gain ($\Delta\mathcal{C}(i) > 0$) at the $n_i$-th eviction event, the eviction volume $K$ must satisfy the following inequality:*

$$K > \frac{6d^2 + d}{n_i \cdot 4Ld} \approx \frac{1.5d}{n_i L} \qquad (10)$$

*Table 1.* Performance comparison of different methods on R1-Llama and R1-Qwen. We report the average Pass@1 and Throughput (TPS) across six benchmarks. "Transformers" denotes the full cache baseline, and "Window" represents the local window baseline.

| Method | AIME24 | | AIME25 | | AMC23 | | GPQA-D | | GK23EN | | MATH500 | | AVERAGE | |
|---|---|---|---|---|---|---|---|---|---|---|---|---|---|---|
| | Pass@1 | TPS | Pass@1 | TPS | Pass@1 | TPS | Pass@1 | TPS | Pass@1 | TPS | Pass@1 | TPS | Pass@1 | TPS |
| *R1-Llama* | | | | | | | | | | | | | | |
| Transformers | 47.3 | 215.1 | 28.6 | 213.9 | 86.5 | 200.6 | 46.4 | 207.9 | 73.1 | 390.9 | 87.5 | 323.4 | 61.6 | 258.6 |
| Window | 18.6 | 447.9 | 14.6 | 441.3 | 59.5 | 409.4 | 37.6 | 408.8 | 47.0 | 622.6 | 58.1 | 590.5 | 39.2 | 486.7 |
| StreamingLLM | 20.6 | 445.8 | 16.6 | 445.7 | 65.0 | 410.9 | 37.8 | 407.4 | 53.4 | 624.6 | 66.1 | 592.1 | 43.3 | 487.7 |
| SepLLM | 30.0 | 448.2 | 20.0 | 445.1 | 71.0 | 414.1 | 39.7 | 406.6 | 61.4 | 635.0 | 74.5 | 600.4 | 49.4 | 491.6 |
| H2O | 38.6 | 426.2 | 22.6 | 423.4 | 82.5 | 396.1 | 41.6 | 381.5 | 67.5 | 601.8 | 82.7 | 573.4 | 55.9 | 467.1 |
| SnapKV | 39.3 | 438.2 | 24.6 | 436.3 | 80.5 | 406.9 | 41.9 | 394.1 | 68.7 | 615.7 | 83.1 | 584.5 | 56.3 | 479.3 |
| R-KV | 44.0 | 437.4 | 26.0 | 434.7 | 86.5 | 409.5 | 44.5 | 394.9 | 71.4 | 622.6 | 85.2 | 589.2 | 59.6 | 481.4 |
| **DYNTS (Ours)** | 49.3 | 444.6 | 29.3 | 443.5 | 87.0 | 412.9 | 46.3 | 397.6 | 72.3 | 631.8 | 87.2 | 608.2 | 61.9 | 489.8 |
| *R1-Qwen* | | | | | | | | | | | | | | |
| Transformers | 52.0 | 357.2 | 35.3 | 354.3 | 87.5 | 376.2 | 49.0 | 349.4 | 77.9 | 593.7 | 91.3 | 517.3 | 65.5 | 424.7 |
| Window | 41.3 | 650.4 | 31.3 | 643.0 | 82.0 | 652.3 | 45.9 | 634.1 | 71.8 | 815.2 | 85.0 | 767.0 | 59.5 | 693.7 |
| StreamingLLM | 42.0 | 655.7 | 29.3 | 648.5 | 85.0 | 657.2 | 45.9 | 631.1 | 71.2 | 824.0 | 85.8 | 786.1 | 59.8 | 700.5 |
| SepLLM | 38.6 | 650.0 | 31.3 | 647.6 | 85.5 | 653.2 | 45.6 | 639.5 | 72.0 | 820.1 | 84.4 | 792.2 | 59.6 | 700.4 |
| H2O | 42.6 | 610.9 | 33.3 | 610.7 | 84.5 | 609.9 | 48.1 | 593.6 | 74.1 | 780.1 | 87.0 | 725.4 | 61.6 | 655.1 |
| SnapKV | 48.6 | 639.6 | 33.3 | 633.1 | 87.5 | 633.2 | 46.5 | 622.0 | 74.9 | 787.4 | 88.2 | 768.7 | 63.2 | 680.7 |
| R-KV | 44.0 | 639.5 | 32.6 | 634.7 | 85.0 | 636.8 | 47.2 | 615.1 | 75.8 | 792.8 | 88.8 | 765.5 | 62.2 | 680.7 |
| **DYNTS (Ours)** | 52.0 | 645.6 | 36.6 | 643.0 | 88.5 | 646.0 | 48.1 | 625.7 | 76.4 | 788.5 | 90.0 | 779.5 | 65.3 | 688.1 |

This inequality provides a theoretical lower bound for the eviction volume $K$. demonstrating that the break-even point is determined by the model's architectural (hidden dimension $d$ and layer count $L$).

# 6. Experiment

This section introduces experimental settings, followed by the results, ablation studies on retanind tokens and hyperparameters, and the Importance Predictor analysis. For more detailed configurations and additional results, please refer to Appendix C and D.

## 6.1. Experimental Setup

**Models and Datasets.** We conduct experiments on two mainstream LRMs: R1-Qwen (DeepSeek-R1-Distill-Qwen-7B) and R1-Llama (DeepSeek-R1-Distill-Llama-8B) (Guo et al., 2025). To evaluate the performance and robustness of our method across diverse tasks, we select five mathematical reasoning datasets of varying difficulty levels—AIME24 (Zhang & Math-AI, 2024), AIME25 (Zhang & Math-AI, 2025), AMC23[3], GK23EN (GAOKAO2023EN)[4], and MATH500 (Hendrycks et al., 2021)—along with the GPQA-D (GPQA-Diamond) (Rein et al., 2023) scientific question-answering dataset as evaluation benchmarks.

**Implementation Details.** (1) *Training Settings*: To train the importance predictor, we sample the model-generated contents with correct answers from the MATH training set and calculate the importance scores of thinking tokens. We freeze the model backbone and optimize only the predictor (3-layer MLP), setting the number of training epochs to 15, the learning rate to 5e-4, and the maximum sequence length to 18,000. (2) *Inference Settings*. Following (Guo et al., 2025), setting the maximum decoding steps to 16,384, the sampling temperature to 0.6, top-$p$ to 0.95, and top-$k$ to 20. We apply budget settings based on task difficulty. For challenging benchmarks (AIME24, AIME25, AMC23, and GPQA-D), we set the budget $B$ to 5,000 with a local window size of 2,000; For simpler tasks, the budget is set to 3,000 with a local window of 1,500 for R1-Qwen and 1,000 for R1-Llama. The token retention ratio in the selection window is set to 0.4 for R1-Qwen and 0.3 for R1-Llama. We generate 5 responses for each problem and report the average Pass@1 as the evaluation metric.

**Baselines.** Our approach focuses on compressing the KV cache by selecting critical tokens. Therefore, we compare our method against the state-of-the-art KV cache compressing approaches. These include StreamingLLM (Xiao et al., 2024), H2O (Zhang et al., 2023), SepLLM (Chen et al., 2024), and SnapKV (Li et al., 2024) (decode-time variant (Liu et al., 2025a)) for LLMs, along with R-KV (Cai et al., 2025) for LRMs. To ensure a fair comparison, all methods were set with the same token overhead and maximum budget. We also report results for standard Transformers and local window methods as evaluation baselines.

---

[3]https://huggingface.co/datasets/math-ai/amc23

[4]https://huggingface.co/datasets/MARIO-Math-Reasoning/Gaokao2023-Math-En

*Table 2.* Ablation study on different token retention strategies in DYNTS, where *w.o.* $Q/T/L$ denotes the removal of Question tokens (Q), critical Thinking tokens (T), and Local window tokens (L), respectively. T-Random and T-Bottom represent strategies that select thinking tokens randomly and the tokens with the bottom-k importance scores, respectively.

| Method | AIME24 | AIME25 | AMC23 | GPQA-D | GK23EN | MATH500 | AVG |
|---|---|---|---|---|---|---|---|
| *R1-Llama* | | | | | | | |
| DynTS | 49.3 | 29.3 | 87.0 | 46.3 | 72.3 | 87.2 | 61.9 |
| *w.o. L* | 40.6 | 23.3 | 86.5 | 46.3 | 72.0 | 85.5 | 59.0 |
| *w.o. Q* | 19.3 | 14.6 | 59.0 | 38.1 | 47.8 | 59.8 | 39.8 |
| *w.o. T* | 44.0 | 27.3 | 85.0 | 44.0 | 71.5 | 85.9 | 59.6 |
| T-Random | 24.6 | 16.0 | 59.5 | 37.4 | 51.7 | 63.9 | 42.2 |
| T-Bottom | 20.6 | 15.3 | 59.0 | 37.3 | 47.3 | 59.5 | 39.8 |
| *R1-Qwen* | | | | | | | |
| DynTS | 52.0 | 36.6 | 88.5 | 48.1 | 76.4 | 90.0 | 65.3 |
| *w.o. L* | 42.0 | 32.0 | 87.5 | 46.3 | 75.2 | 87.0 | 61.6 |
| *w.o. Q* | 46.0 | 36.0 | 86.0 | 43.9 | 75.1 | 89.0 | 62.6 |
| *w.o. T* | 47.3 | 34.6 | 85.5 | 49.1 | 75.1 | 89.2 | 63.5 |
| T-Random | 46.0 | 32.6 | 84.5 | 47.5 | 73.8 | 86.9 | 61.9 |
| T-Bottom | 38.0 | 30.0 | 80.0 | 44.3 | 69.8 | 83.3 | 57.6 |

## 6.2. Main Results

**Reasoning Accuracy.** As shown in Table 1, our proposed DYNTS consistently outperforms all other KV cache eviction baselines. On R1-Llama and R1-Qwen, DYNTS achieves an average accuracy of 61.9% and 65.3%, respectively, significantly surpassing the runner-up methods R-KV (59.6%) and SnapKV (63.2%). Notably, the overall reasoning capability of DYNTS is on par with the Full Cache Transformers baseline (61.9% vs. 61.6% on R1-Llama, 65.3% vs. 65.5% on R1-Llama). Furthermore, we conduct a statistical analysis of the reasoning accuracy. Detailed results are provided in Appendix D.

**Inference Efficiency.** Referring to Table 1, DYNTS achieves $1.9\times$ and $1.6\times$ speedup compared to standard Transformers on R1-Llama and R1-Qwen, respectively, across all benchmarks. While DYNTS maintains throughput comparable to other KV Cache compression methods. Further observing Figure 5, as the generated sequence length grows, standard Transformers suffer from linear accumulation in both memory footprint and compute overhead (GFLOPs), leading to continuous throughput degradation. In contrast, DYNTS effectively bounds resource consumption. The distinctive sawtooth pattern illustrates our periodic compression mechanism, where the inflection points correspond to the execution of KV Cache Selection to evict the KV pairs of non-essential thinking tokens. Consequently, the efficiency advantage escalates as the decoding step increases, DYNTS achieves a peak speedup of $4.51\times$, compresses the memory footprint to $0.19\times$, and reduces the compute overhead to $0.52\times$ compared to the full-cache baseline. The zoom-in view reveals that the computational cost drops below the baseline immediately after the first KV cache eviction. This illustrates that our experimental

settings are rational, as they satisfy the break-even condition ($K = 900 \geq \frac{1.5d}{n_i L} = 192$) outlined in Corollary 5.2.

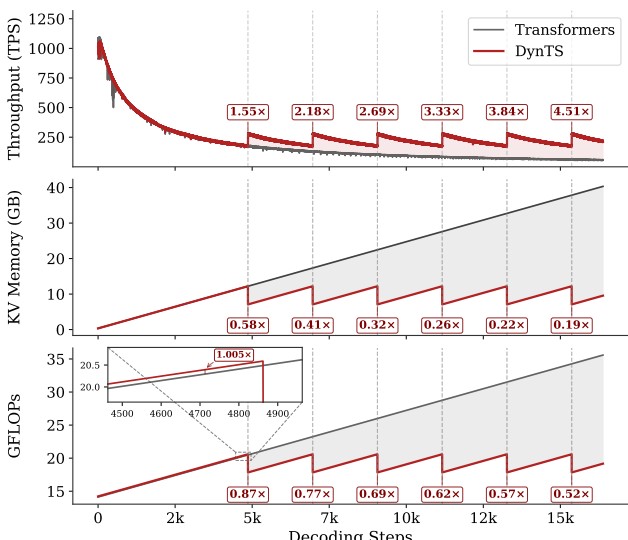

*Figure 5.* Real-time throughput, memory, and compute overhead tracking over total decoding step. The inflection points in the sawtooth correspond to the steps where DYNTS executes KV Cache Selection.

## 6.3. Ablation Study

**Impact of Retained Token.** As shown in Tab. 2, the full DYNTS method outperforms all ablated variants, achieving the highest average accuracy on both R1-Llama (61.9%) and R1-Qwen (65.3%). This demonstrates that all retained tokens of DYNTS are critical for the model to output the correct final answer. Moreover, we observe that the strategy for selecting thinking tokens plays a critical role in the model's reasoning performance. When some redundant tokens are retained (T-Random and T-Bottom strategies), there is a significant performance drop compared to completely removing thinking tokens ($59.6\% \rightarrow 39.8\%$ on R1-Llama and $63.5\% \rightarrow 57.6\%$ on R1-Qwen). This finding demonstrates the effectiveness of our Importance Predictor to identify critical tokens. It also explains why existing KV cache compression methods hurt model performance: inadvertently retaining redundant tokens. Finally, the local window is crucial for preserving local linguistic coherence, which contributes to stable model performance.

**Local Window & Retention Ratio.** As shown in Fig. 6, we report the model's reasoning performance across different configurations. The performance improves with a larger local window and a higher retention ratio within a reasonable range. These two settings respectively ensure local contextual coherence and an adequate number of thinking tokens. Setting either to overly small values leads to pronounced performance degradation. However, excessively

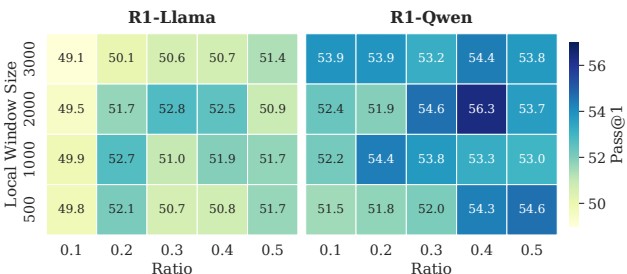

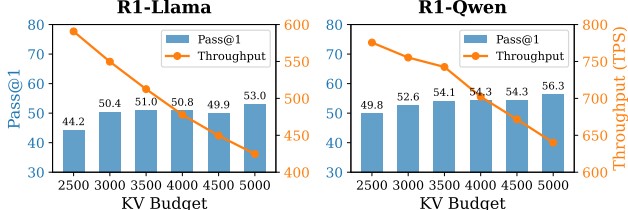

Figure 8. Accuracy and throughput across varying KV budgets.

*Figure 6.* The accuracy of R1-Llama and R1-Qwen across different local window sizes and selection window retention ratios.

large values introduce a higher proportion of non-essential tokens, which in turn negatively impacts model performance. Empirically, a local window size of approximately 2,000 and a retention ratio of 0.3–0.4 yield optimal performance. We further observe that R1-Qwen is particularly sensitive to the local window size. This may be caused by the Dual Chunk Attention introduced during the long-context pretraining stage (Yang et al., 2025), which biases attention toward tokens within the local window.

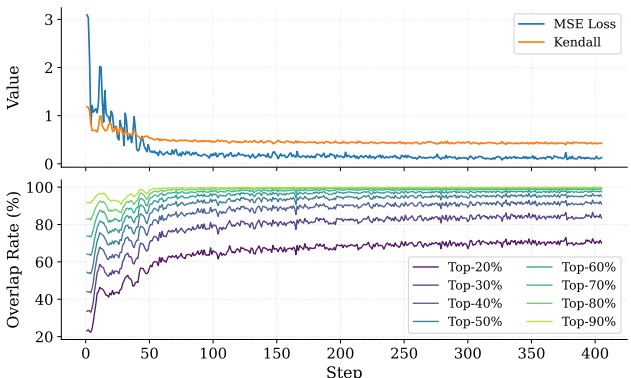

*Figure 7.* The top panel illustrates the convergence of MSE Loss and the Kendall rank correlation coefficient over training steps. The bottom panel tracks the overlap rate of the top-20% ground-truth tokens within the top-$p$% ($p \in [20, 90]$) predicted tokens.

**Budget.** We report the model's reasoning performance and throughput in different budget settings in Fig. 8. As expected, as the KV budget increases, the accuracy of R1-Llama and R1-Qwen improves, and the throughput decreases. At the maximum evaluated budget of 5000, DYNTS delivers its strongest reasoning results (53.0% for R1-Llama and 56.3% for R1-Qwen), minimizing the performance gap with the full-cache baseline.

### 6.4. Analysis of Importance Predictor

To validate that the Importance Predictor effectively learns the ground-truth thinking token importance scores, we report the MSE Loss and the Kendall rank correlation coefficient (Abdi, 2007) in the top panel of Fig. 7. As the number

of training steps increases, both metrics exhibit clear convergence. The MSE loss demonstrates that the predictor can fit the true importance scores. The Kendall coefficient measures the consistency of rankings between the ground-truth importance scores and the predicted values. This result indicates that the predictor successfully captures each thinking token's importance to the answer. Furthermore, we analyze the overlap rate of predicted critical thinking tokens, as shown in the bottom panel of Fig. 7. Notably, at the end of training, the overlap rate of critical tokens within the top 30% of the predicted tokens exceeds 80%. This confirms that the Importance Predictor in DYNTS effectively identifies the most pivotal tokens, ensuring the retention of essential thinking tokens even at high compression rates.

## 7. Related Work

Recent works on KV cache compression have primarily focused on classical LLMs, applying eviction strategies based on attention scores or heuristic rules. One line of work addresses long-context pruning at the prefill stage. Such as SnapKV (Li et al., 2024), PyramidKV (Cai et al., 2024), and AdaKV (Feng et al., 2024). However, they are ill-suited for the inference scenarios of LRMs, which have short prefill tokens followed by long decoding steps. Furthermore, several strategies have been proposed specifically for the decoding phase. For instance, H2O (Zhang et al., 2023) leverages accumulated attention scores, StreamingLLM (Xiao et al., 2024) retains attention sinks and recent tokens, and SepLLM (Chen et al., 2024) preserves only the separator tokens. More recently, targeting LRMs, (Cai et al., 2025) introduced RKV, which adds a similarity-based metric to evict redundant tokens, while RLKV (Du et al., 2025) utilizes reinforcement learning to retain critical reasoning heads. However, these methods fail to accurately assess the contribution of intermediate tokens to the final answer. Consequently, they risk erroneously evicting decision-critical tokens, compromising the model's reasoning performance.

## 8. Conclusion and Discussion

In this work, we investigated the relationship between the reasoning traces and their final answers in LRMs. Our analysis revealed a Pareto Principle in LRMs: only the

decision-critical thinking tokens ($20\% \sim 30\%$ in the reasoning traces) steer the model toward the final answer. Building on this insight, we proposed DYNTS, a novel KV cache compression method. Departing from current strategies that rely on local attention scores for eviction, DYNTS introduces a learnable Importance Predictor to predict the contribution of the current token to the final answer. Based on the predicted score, DYNTS retains pivotal KV cache. Empirical results on six datasets confirm that DYNTS outperforms other SOTA baselines. We also discuss the limitations of DYNTS and outline potential directions for future improvement. Please refer to Appendix E for details.

## Acknowledgements

This research was supported by the National Natural Science Foundation of China under Grant No. 62302441. This work was also supported by the Key Research and Development Program Project of Ningbo, Grant No. 2025Z029. The author gratefully acknowledges the support of Zhejiang University Education Foundation and Qizhen Scholar Foundation. Additional support was provided by the Information Technology Center of Zhejiang University and the Supercomputing Center of Hangzhou City University.

## Impact Statement

This paper presents work aimed at advancing the field of KV cache compression. There are many potential societal consequences of our work, none of which we feel must be specifically highlighted here. The primary impact of this research is to improve the memory and computational efficiency of LRM's inference. By reducing memory requirements, our method helps lower the barrier to deploying powerful models on resource-constrained edge devices. We believe our work does not introduce specific ethical or societal risks beyond the general considerations inherent to advancing generative AI.

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

# A. Proof of Cumulative Computational Gain

**Definition A.1** (Predictor Overhead). Let $\mathcal{M}_{\text{base}}$ be the vanilla LRM with $L$ layers and a hidden dimension $d$, and $\mathcal{M}_{\text{opt}}$ be the LRM with Importance Predictor. The predictor is defined as a three-layer linear MLP with dimensions $d \rightarrow m_1 \rightarrow m_2 \rightarrow m_3$. The computational cost per decode step is:

$$\mathcal{C}_{\text{mlp}} = 2(d \cdot m_1 + m_1 \cdot m_2 + m_2 \cdot m_3). \tag{11}$$

Setting $m_1 = 2d$, $m_2 = d/2$, and $m_3 = 1$ yields:

$$\mathcal{C}_{\text{mlp}} = 6d^2 + d \tag{12}$$

**Definition A.2** (Effective KV Cache Length). Let $M$ denote the length of the prefill sequence. At decode step $i$, when the effective cache length reaches the budget $B$, DYNTS performs KV Cache Selection to evict $K$ redundant tokens. The effective KV cache length $S_i$ for the base and optimized models is given by:

$$S_i^{\text{base}} = M + i, \quad S_i^{\text{opt}} = M + i - n_i \cdot K, \tag{13}$$

where $n_i = \max\left(0, \left\lfloor \frac{(M+i)-B}{K} \right\rfloor + 1\right)$ denotes the count of cache eviction event at step $i$.

**Definition A.3** (LLM Overhead). The computational overhead per step for a decoder-only transformer is composed of a static component $\mathcal{C}_{\text{static}}$ (independent of sequence length) and a dynamic attention component $\mathcal{C}_{\text{attn}}$ (linearly dependent on effective cache length). The static cost $\mathcal{C}_{\text{static}}$ for the backbone remains identical for both models. The self-attention cost for a single layer is $4 \cdot d \cdot S_i$ (counting $Q \cdot K^\top$ and Softmax $\cdot V$). Across $L$ layers:

$$\mathcal{C}_{\text{attn}}(S_i) = 4 \cdot L \cdot d \cdot S_i \tag{14}$$

*Proof: Computational Gain.* Let $\Delta\mathcal{C}(i)$ be the reduction in FLOPs achieved by DYNTS at decoding step $i$, which is defined as $\text{FLOPs}(\mathcal{M}_{\text{base}}(i)) - \text{FLOPs}(\mathcal{M}_{\text{opt}}(i))$:

$$\Delta\mathcal{C}(i) = \left[\mathcal{C}_{\text{static}}(i) + \mathcal{C}_{\text{attn}}(S_i^{\text{base}})\right] - \left[\mathcal{C}_{\text{static}}(i) + \mathcal{C}_{\text{mlp}} + \mathcal{C}_{\text{attn}}(S_i^{\text{opt}})\right]. \tag{15}$$

Eliminating the static term $\mathcal{C}_{\text{static}}$

$$\Delta\mathcal{C}(i) = \mathcal{C}_{\text{attn}}(S_i^{\text{base}}) - \mathcal{C}_{\text{attn}}(S_i^{\text{opt}}) - \mathcal{C}_{\text{mlp}}. \tag{16}$$

Substituting $S_i^{\text{base}}$, $S_i^{\text{opt}}$ and $\mathcal{C}_{\text{mlp}}$:

$$\Delta\mathcal{C}(i) = 4 \cdot L \cdot d \cdot (M + i) - 4 \cdot L \cdot d \cdot (M + i - n_i \cdot K) - \mathcal{C}_{\text{mlp}} \tag{17}$$

$$= \underbrace{n_i \cdot 4LdK}_{\text{Eviction Saving}} - \underbrace{(6d^2 + d)}_{\text{Predictor Overhead}}. \tag{18}$$

This completes the proof. $\square$

$\square$

# B. Empirical Analysis and Observations

## B.1. Implementation Details

To calculate the importance of each thinking token to the final answer sequence, we first utilized vLLM (Kwon et al., 2023) to generate the complete reasoning trace. Following (Guo et al., 2025), we set the temperature to 0.6, top-p to 0.95, top-k to 20, and max length to 16,384. To ensure sequence completeness, we sampled 5 times for each question and filtered out samples with incomplete inference traces. Then, we fed the full sequence into the model for a single forward pass to extract the submatrices of attention weight, corresponding to the answer and thinking tokens. Finally, we aggregate the matrices across all layers and heads, and sum along the answer dimension (rows). The aggregated 1D vector is the importance score of each thinking token.

Based on the calculated importance scores, we employ three selection strategies to retain critical thinking tokens: top-$p\%$, bottom-$p\%$, and random sampling, where $p \in [2, 4, 6, 8, 10, 20, 30, 40, 50]$. The retained tokens are concatenated with the original question to form the input sequence. The input sequence is processed by vLLM over 5 independent runs using the aforementioned configuration. We report the average Pass@1 across these results as the final accuracy.

## B.2. Ratio of Content Words

To investigate the distinctions of thinking tokens with varying importance scores, we employed spaCy to analyze the Part-of-Speech (POS) tags of each token. Specifically, we heuristically categorized nouns, verbs, adjectives, adverbs, and proper nouns as *Content Words* carrying substantive meaning, while treating other POS tags as *Function Words* with limited semantic information. The thinking tokens were sorted by importance score and then partitioned into ten equal parts. We report the ratio of Content Words and Function Words within each part in Fig 9. The tokens with higher importance scores exhibit a significantly higher proportion of content words, suggesting that they encode the core semantic meaning. Conversely, tokens with lower scores are predominantly function words, which primarily serve as syntactic scaffolding or intermediate states to maintain sequence coherence. Consequently, once the full sentence is generated, removing these low-importance tokens has a negligible impact on overall comprehension.

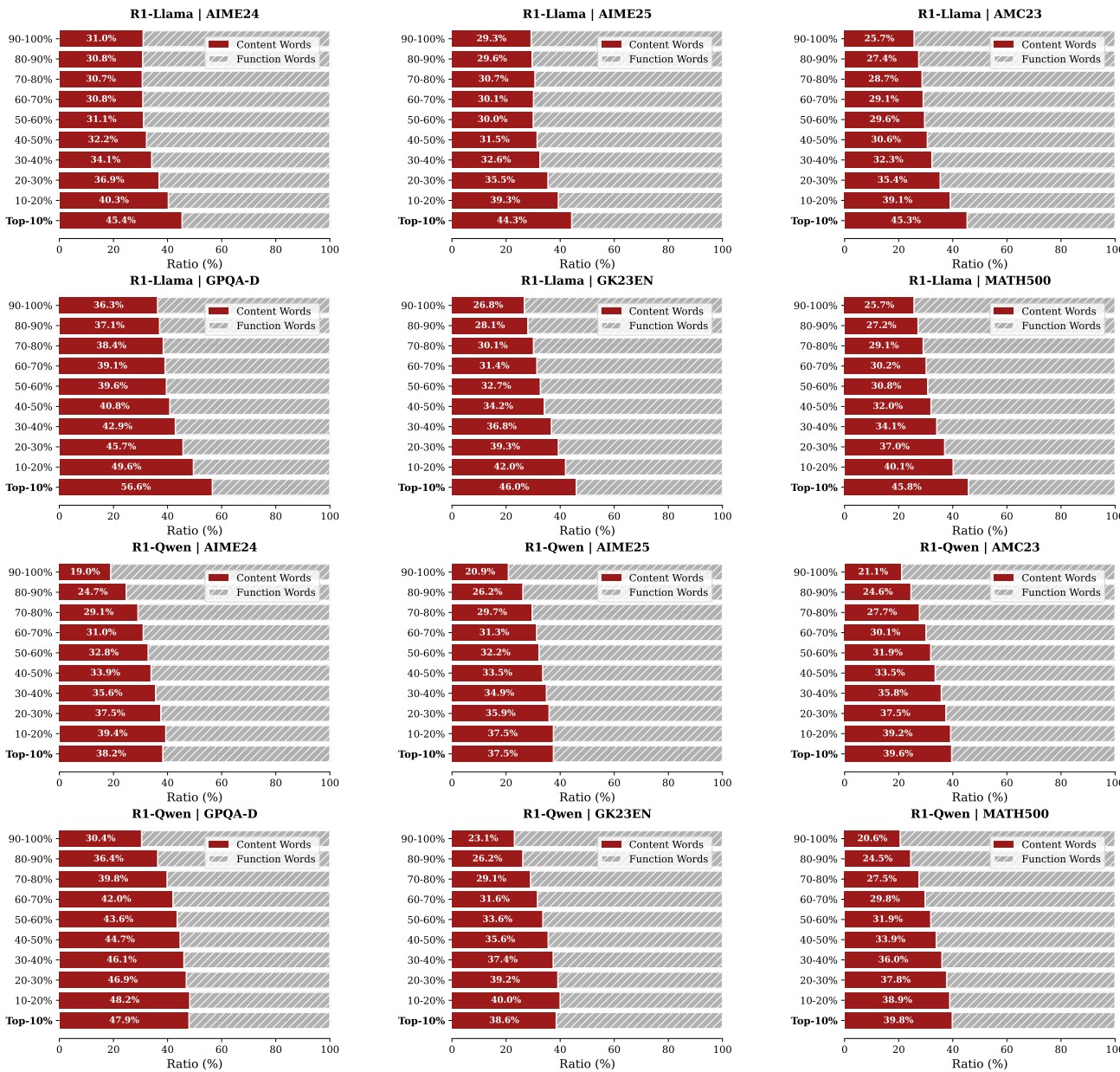

*Figure 9.* Proportion of content words versus function words in thinking tokens. Bars represent deciles sorted by importance score; e.g., the bottom bar indicates the ratio for the top-10% most important tokens.

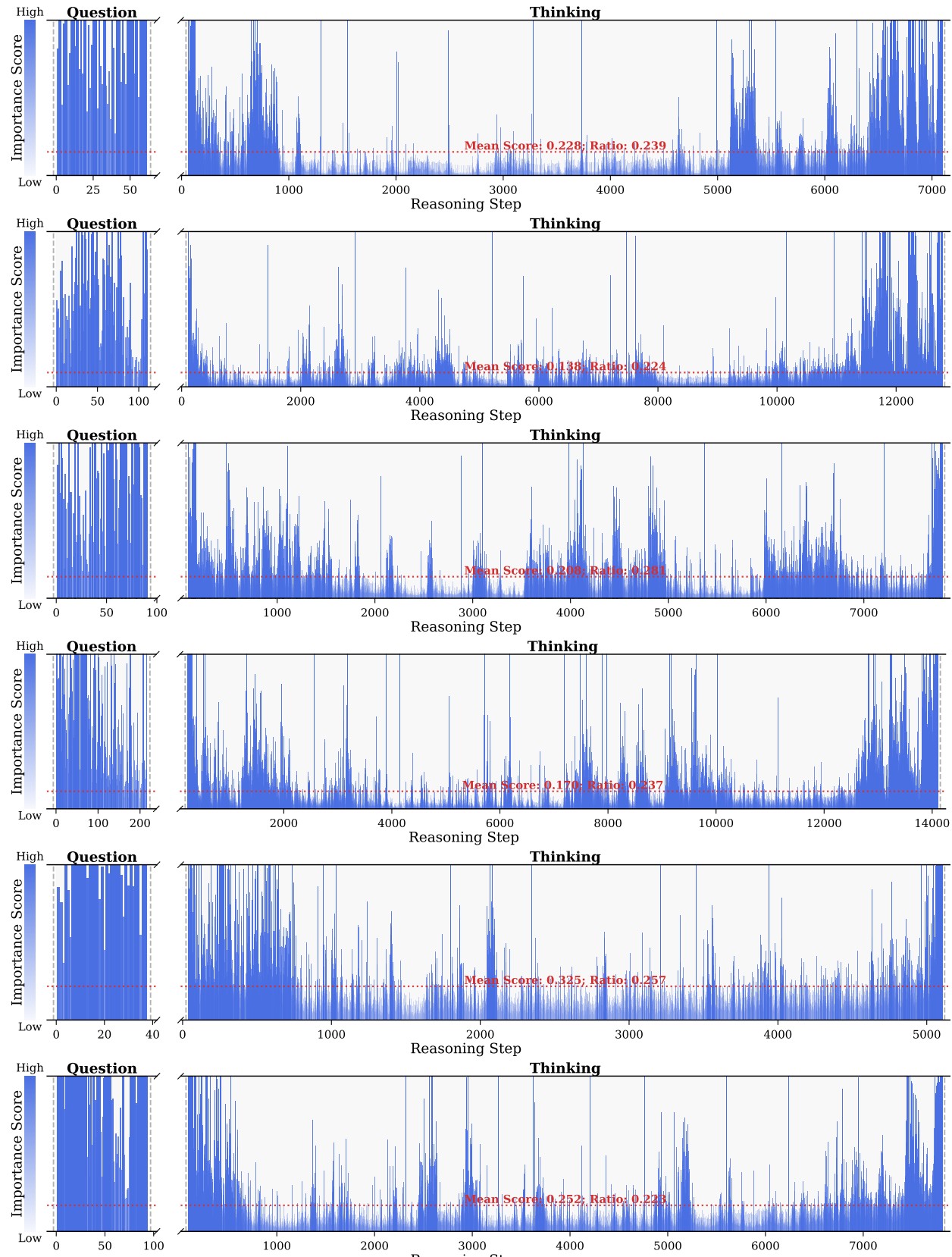

*Figure 10.* Visualization of token importance scores for question and thinking tokens within DeepSeek-R1-Distill-Llama-8B reasoning traces across six samples from AIME24, AIME25, AMC23, GPQA-D, GAOKAO2023EN, and MATH500.

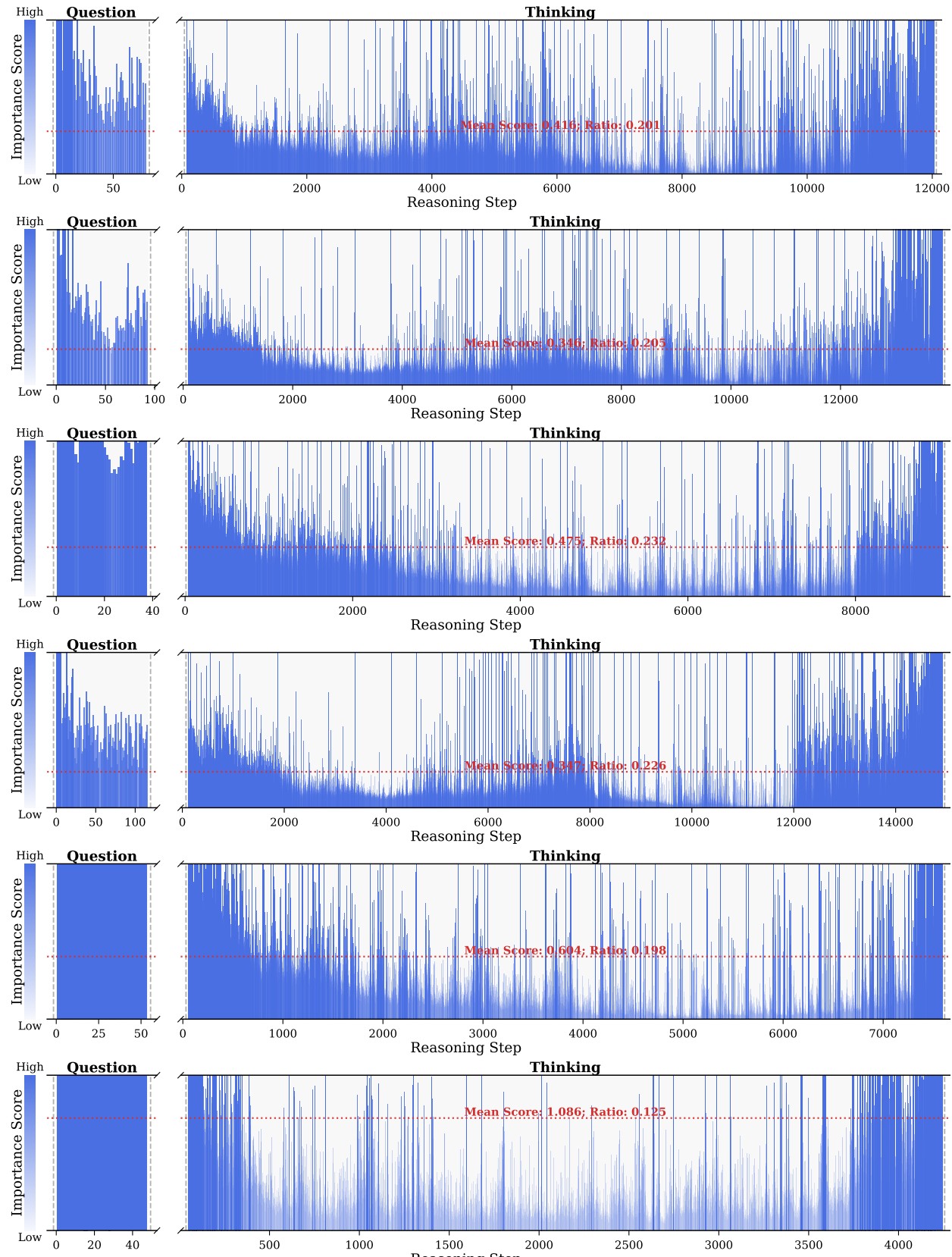

*Figure 11.* Visualization of token importance scores for question and thinking tokens within DeepSeek-R1-Distill-Qwen-7B reasoning traces across six samples from AIME24, AIME25, AMC23, GPQA-D, GAOKAO2023EN, and MATH500.

## B.3. Additional Results

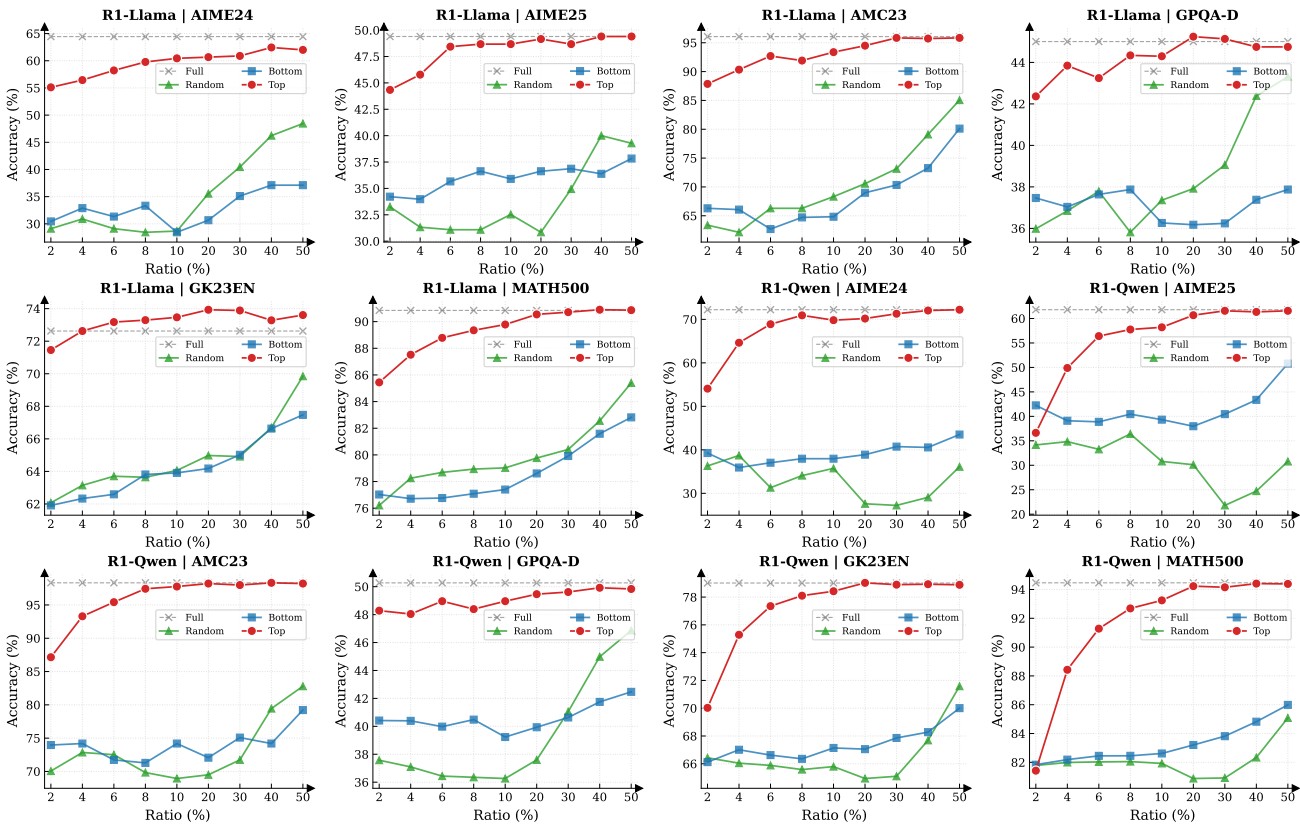

*Figure 12.* Reasoning performance trends of Llama and Qwen models across six datasets as a function of thinking token retention ratio.

We present the evaluation results of R1-Llama (DeepSeek-R1-Distill-Llama-8B) and R1-Qwen (DeepSeek-R1-Distill-Qwen-7B) across the AIME24, AIME25, AMC23, GPQA-Diamond, GK23EN, and MATH-500 datasets. Fig. 10 and 11 illustrate the variation of token importance scores over decoding steps for representative samples from each dataset. Additionally, Fig. 12 demonstrates the reasoning performance under different thinking token retention strategies and retention ratios.

# C. Detailed Experimental Setup

## C.1. Device and Environment

All experiments were conducted on a server equipped with 8 NVIDIA H800 GPUs. We utilize the Hugging Face `transformers` library and `vLLM` as the primary inference engine, and employ the `veRL` framework to optimize the parameters of the importance predictor.

## C.2. Training of Importance Predictor

We generate 5 diverse reasoning traces for each question in the MATH training dataset. Only the traces leading to the correct answer are retained as training data. This process yields 7,142 and 7,064 valid samples for R1-Qwen and R1-Llama, respectively. The Importance Predictor is implemented as an MLP with dimensions of $(3584 \rightarrow 7168 \rightarrow 1792 \rightarrow 1)$ and $(4096 \rightarrow 8192 \rightarrow 2048 \rightarrow 1)$ for R1-Qwen and R1-Llama. We conduct the training using the `veRL` framework for a total of 15 epochs. The global batch size is set to 256, with a micro-batch size of 4 for the gradient accumulation step. Optimization is performed using AdamW optimizer with $\beta_1 = 0.9$, $\beta_2 = 0.95$, and a weight decay of $0.01$. The learning rate is initialized at $5 \times 10^{-4}$ with a cosine decay schedule, and the gradient clipping threshold is set to $1.0$.

## C.3. Inference Setting

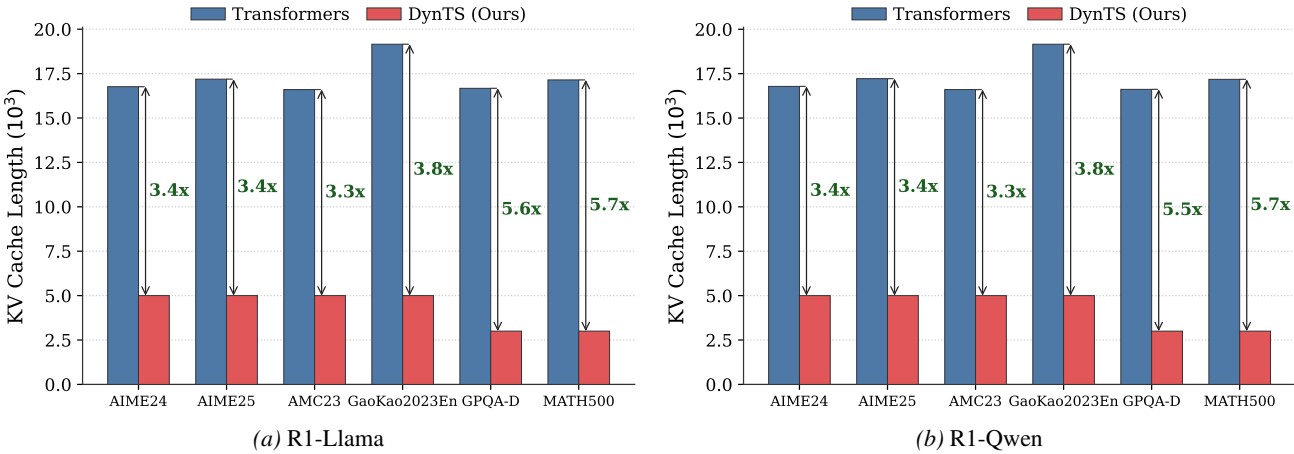

*Figure 13.* Comparison of average KV Cache length between standard Transformers and DYNTS across six benchmarks. The arrows and annotations indicate the compression ratio achieved by our method on each dataset.

We implement DYNTS and all baseline methods using the Hugging Face `transformers` library for KV cache compression. To ensure fairness, we use the effective KV cache length as the compression signal. Whenever the cache size reaches the predefined budget, all methods are restricted to retain an identical number of KV pairs. For SnapKV, H2O, and R-KV, we set identical local window sizes and retention ratios to those of our methods. For SepLLM, we preserve the separator tokens and evict the earliest generated non-separator tokens until the total cache length matches ours. For StreamingLLM, we set the same sink token size, following (Xiao et al., 2024). We set the number of parallel generated sequences to 20. The generation hyperparameters are configured as follows: temperature $T = 0.6$, top-$p = 0.95$, top-$k = 20$, and a maximum new token limit of 16,384. We conduct 5 independent sampling runs for all datasets. Ablation studies on the local window, retention ratio, and budget were conducted across four challenging benchmarks (AIME24, AIME25, AMC23, and GPQA-D) with the same other configurations to verify effectiveness. Fig. 6 and 8 report the mean results across these datasets.

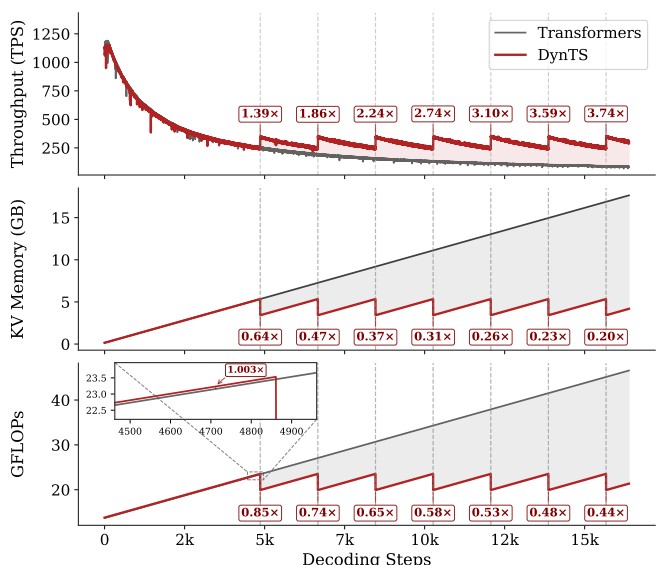

*Figure 14.* Real-time throughput, memory, and compute overhead tracking for R1-Qwen over total decoding steps. The results exhibit a trend consistent with R1-Llama, confirming the scalability of DYNTS across different model architectures.

## D. Additional Results

**Statistical Analysis.** We conduct statistical analysis and report the $95\%$ CI (Confidence Intervals) for all main results, as shown in Table 3 and 4. The CI provides a more reliable comparison for different methods. Across both R1-Llama and R1-Qwen, DynTS shows CIs are largely comparable to those of the full-cache Transformers baseline on most benchmarks, suggesting that DynTS maintains on-par reasoning performance. Compared with existing compressed-cache baselines, DynTS exhibits higher CI across benchmarks. Overall, the $95\%$ CI results demonstrate that DynTS achieves reliable and robust performance across different benchmarks.

**Generalizability.** We further evaluated our method on the code-reasoning benchmark HumanEval using the predictor trained only on the MATH training set. The results are reported in the Table 5. Notably, even without fine-tuning on code tasks, our method incurs only $0.8\%$ and $1.2\%$ performance drops compared with the full-cache baseline in R1-Qwen and R1-Llama, respectively. This suggests that our method generalizes well and can be applied to other reasoning tasks.

*Table 3.* Confidence intervals of R1-Llama across all benchmarks. Each entry denotes the lower and upper bounds of the reasoning performance for the corresponding method.

| Method | AIME24 | AIME25 | AMC23 | GK2023EN | GPQA_D | MATH500 |
|---|---|---|---|---|---|---|
| Transformers | [37.9, 56.7] | [23.9, 33.3] | [83.0, 90.0] | [70.9, 75.3] | [41.7, 51.1] | [86.0, 89.0] |
| Window | [12.4, 24.8] | [10.9, 18.3] | [54.9, 64.1] | [44.3, 49.7] | [35.4, 39.8] | [57.0, 59.2] |
| StreamingLLM | [16.1, 25.1] | [13.7, 19.5] | [60.1, 69.9] | [51.4, 55.4] | [36.7, 38.9] | [65.2, 67.0] |
| SepLLM | [18.7, 41.3] | [17.1, 22.9] | [67.5, 74.5] | [59.6, 63.2] | [36.9, 42.5] | [72.9, 76.1] |
| H2O | [31.7, 45.5] | [20.8, 24.4] | [76.3, 88.7] | [64.5, 70.5] | [37.6, 45.6] | [81.1, 84.3] |
| SnapKV | [31.9, 46.7] | [18.4, 30.8] | [73.8, 87.2] | [66.8, 70.6] | [39.1, 44.7] | [81.6, 84.6] |
| R-KV | [37.9, 50.1] | [22.6, 29.4] | [79.8, 93.2] | [68.4, 74.4] | [42.2, 46.8] | [84.4, 86.0] |
| DynTS | [41.4, 57.2] | [25.9, 32.7] | [84.5, 89.5] | [70.0, 74.6] | [42.8, 49.8] | [86.1, 88.1] |

*Table 4.* Confidence intervals of R1-Qwen across all benchmarks. Each entry denotes the lower and upper bounds of the reasoning performance for the corresponding method.

| Method | AIME24 | AIME25 | AMC23 | GK2023EN | GPQA_D | MATH500 |
|---|---|---|---|---|---|---|
| Transformers | [41.7, 62.3] | [31.6, 39.0] | [83.2, 91.8] | [76.5, 79.3] | [46.6, 51.4] | [90.4, 92.2] |
| Window | [35.1, 47.5] | [29.1, 33.5] | [76.1, 87.9] | [70.3, 73.3] | [43.0, 48.8] | [84.4, 85.6] |
| StreamingLLM | [35.1, 48.9] | [25.9, 32.7] | [81.2, 88.8] | [70.0, 72.4] | [42.6, 49.2] | [85.2, 86.4] |
| SepLLM | [30.6, 46.6] | [24.4, 38.2] | [81.5, 89.5] | [70.2, 73.8] | [43.8, 47.4] | [83.7, 85.1] |
| H2O | [39.2, 46.0] | [27.5, 39.1] | [80.5, 88.5] | [72.8, 75.4] | [45.8, 50.4] | [86.1, 87.9] |
| SnapKV | [43.1, 54.1] | [28.3, 38.3] | [84.4, 90.6] | [73.0, 76.8] | [44.9, 48.1] | [86.7, 89.7] |
| R-KV | [34.6, 53.4] | [27.3, 37.9] | [79.7, 90.3] | [73.9, 77.7] | [43.1, 51.3] | [87.9, 89.7] |
| DynTS | [47.3, 56.7] | [31.6, 41.6] | [85.0, 92.0] | [75.4, 77.4] | [45.2, 51.0] | [89.2, 90.8] |

**Inference Efficiency on R1-Qwen.** Complementing the efficiency analysis of R1-Llama presented in the main text, Figure 14 illustrates the real-time throughput, memory footprint, and computational overhead for R1-Qwen. Consistent with previous observations, DYNTS exhibits significant scalability advantages over the standard Transformer baseline as the sequence length increases, achieving a peak throughput speedup of $3.74\times$ while compressing the memory footprint to $0.20\times$ and reducing the cumulative computational cost (GFLOPs) to $0.44\times$ after the last KV cache selection step. The recurrence of the characteristic sawtooth pattern further validates the robustness of our periodic KV Cache Selection mechanism, demonstrating that it effectively bounds resource accumulation and delivers substantial efficiency gains across diverse LRM architectures by continuously evicting non-essential thinking tokens.

*Table 5.* Performance comparison on HumanEval dataset.

| Method | R1-Qwen | R1-Llma |
|---|---|---|
| *Transformers* | *89.7* | *83.8* |
| **DynTS (Our)** | **88.9** | **82.6** |
| StreamingLLM | 78.5 | 54.0 |
| SepLLM | 76.9 | 67.5 |
| H2O | 86.2 | 81.5 |
| SnapKV | 85.8 | 82.3 |
| R-KV | 86.7 | 81.2 |

**KV Cache Compression Ratio.** Figure 13 explicitly visualizes the reduction in KV Cache length achieved by DYNTS across diverse reasoning tasks. By dynamically filtering out non-essential thinking tokens, our method drastically reduces the memory footprint compared to the full-cache Transformers baseline. For instance, on the MATH500 benchmark, DYNTS achieves an impressive compression ratio of up to $5.7\times$, reducing the average cache length from over 17,000 tokens to the constrained budget of 3,000. These results directly explain the memory and throughput advantages reported in the efficiency analysis, confirming that DYNTS successfully maintains high reasoning accuracy with a fraction of the memory cost.

**Importance Predictor Analysis for R1-Qwen.** Complementing the findings on R1-Llama, Figure 15 depicts the learning trajectory of the Importance Predictor for the R1-Qwen model. The training process exhibits a similar convergence pattern: the MSE loss rapidly decreases and stabilizes, while the Kendall rank correlation coefficient steadily improves, indicating that the simple MLP architecture effectively captures the importance ranking of thinking tokens in R1-Qwen. Furthermore, the bottom panel highlights the high overlap rate between the predicted and ground-truth critical tokens; notably, the overlap rate for the top-40% of tokens exceeds 80% after approximately 200 steps. This high alignment confirms that the Importance

Predictor can accurately identify pivotal tokens within R1-Qwen's reasoning process, providing a reliable basis for the subsequent KV cache compression.

**Budget Impact Analysis over Benchmarks.** Figures 16 illustrate the granular impact of KV budget constraints on reasoning performance and system throughput. Focusing on R1-Llama, we observe a consistent trade-off across all datasets: increasing the KV budget significantly boosts reasoning accuracy at the cost of linearly decreasing throughput. Specifically, on the challenging AIME24 benchmark, expanding the budget from 2,500 to 5,000 tokens improves Pass@1 accuracy from $40.0\%$ to $49.3\%$, while the throughput decreases from ∼600 to ∼445 tokens/s. This suggests that while a tighter budget accelerates inference, a larger budget is essential for solving complex problems requiring extensive context retention. Experimental results on R1-Qwen exhibit a highly similar trend, confirming that the performance characteristics of DYNTS are model-agnostic. Overall, our method allows users to flexibly balance efficiency and accuracy based on specific deployment requirements.

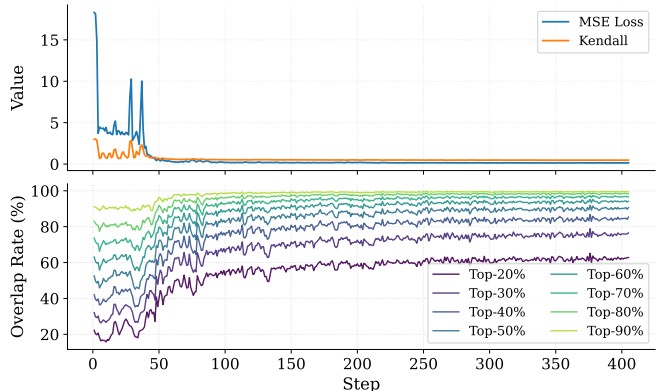

*Figure 15.* Training dynamics of the Importance Predictor on R1-Qwen. The top panel displays the convergence of MSE Loss and Kendall correlation, while the bottom panel shows the overlap rate of the top-20% ground-truth tokens within the top-$p\%(p \in [20, 90])$ predicted tokens across training steps.

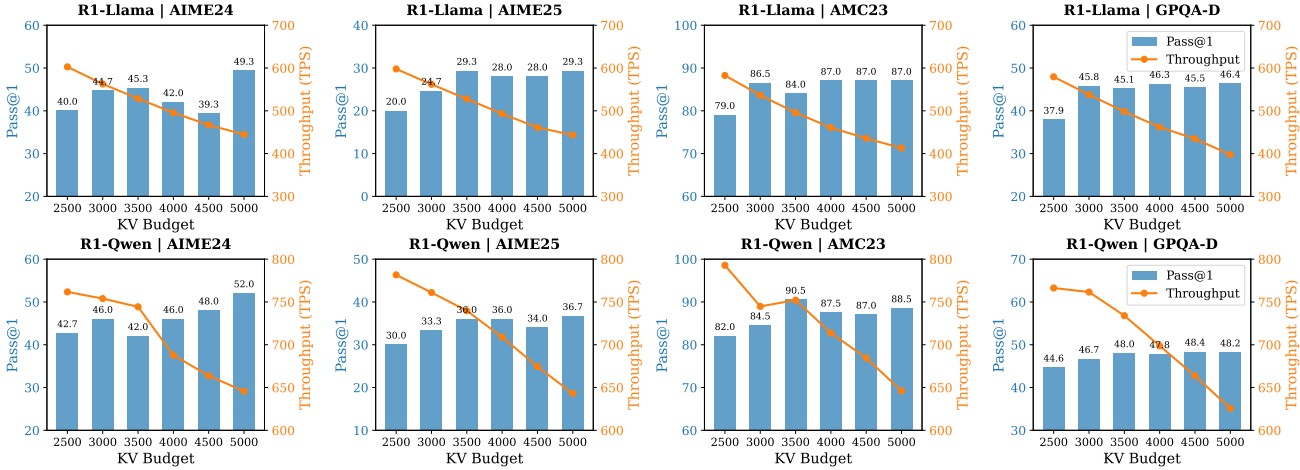

*Figure 16.* Impact of budget on Pass@1 and throughput for R1-Llama (top) and R1-Qwen (bottom) across AIME24, AIME25, AMC23, and GPQA-D datasets. The blue bars represent accuracy (left y-axis), and the orange lines represent throughput (right y-axis).

**Local Window and Retention Ratio Analysis over Benchmarks.** Figures 17 illustrate the sensitivity of model performance to variations in Local Window Size and the Retention Ratio of the Selection Window. A moderate local window (e.g., 1000–2000) typically yields optimal results, suggesting that recent context saturation is reached relatively. Furthermore, we observe that the retention ratio between $0.3$ and $0.4$ across most benchmarks (e.g., AIME24, GPQA), where the model effectively balances and reasoning performance. Whereas lower ratios (e.g., $0.1$) consistently degrade accuracy due to excessive information loss.

# E. Limitations and Future Work

Currently, DYNTS is implemented based on the `transformers` library, and we are actively working on deploying it to other inference frameworks such as `vLLM` and `SGLang`. Additionally, our current training data focuses on mathematical reasoning, which may limit performance in other domains like coding or abstract reasoning. In the future, we plan to expand data diversity to adapt to a broader range of reasoning tasks. Moreover, constrained by computational resources, we used a relatively small dataset (∼ $7,000$ samples) for training. This dataset's scale limits us to optimizing only the

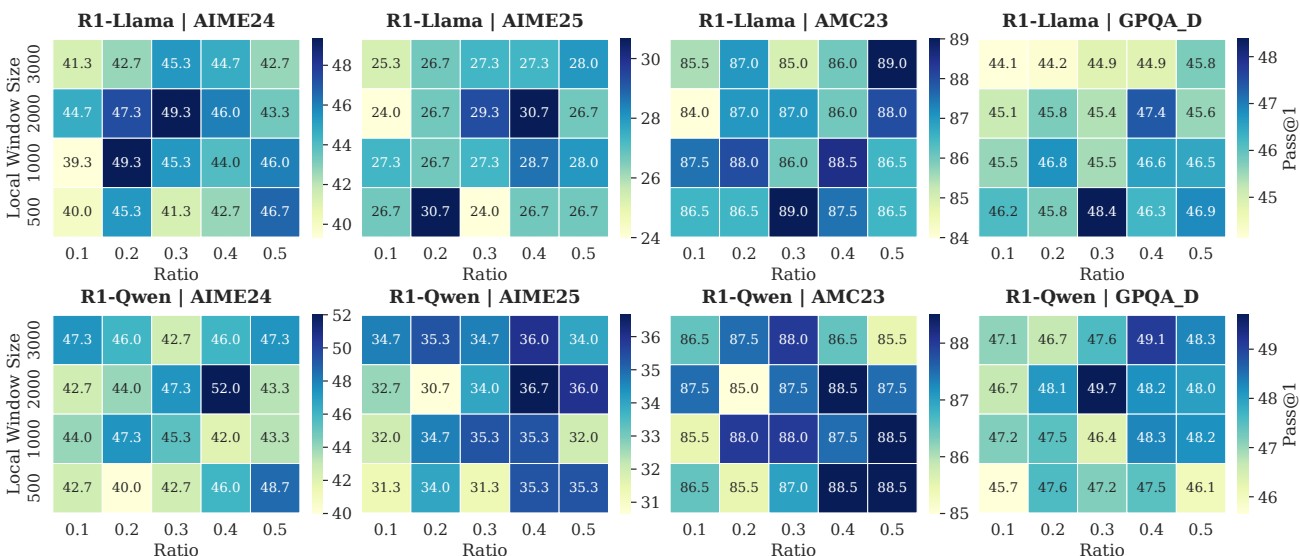

*Figure 17.* Impact of different local window sizes and retention ratios of section window

importance predictor's parameters, since optimizing all parameters on the small dataset may compromise the model's original generalization capabilities. This constraint may hinder the full potential of DYNTS. Future work can focus on scaling up the dataset and jointly optimizing both the backbone and the predictor to elicit stronger capabilities.

