# OpenReview forum: "Dynamic Thinking-Token Selection for Efficient Reasoning in Large Reasoning Models"
_ICML.cc/2026/Conference — ICML 2026 regular_

### Official Review · Reviewer_CzcE · 2026-03-12

**Soundness:** 3
**Presentation:** 3
**Significance:** 3
**Originality:** 2
**Overall Recommendation:** 5
**Confidence:** 4

**Summary:**

This paper addresses the memory and computational bottleneck of Long Reasoning Models (LRMs) caused by their extended thinking traces. The authors first observe that thinking tokens exhibit a sparse importance distribution — only ~20-30% of tokens receive significant attention from the final answer, following a "Pareto Principle." Based on this, they propose DYNTS (Dynamic Thinking-Token Selection), which attaches a lightweight MLP importance predictor to the final hidden layer of the LRM to predict each thinking token's importance on-the-fly. A dual-window KV cache selection strategy (Selection Window for high-importance tokens + Local Window for recent context) then retains only critical KV entries when the cache reaches a budget limit. Evaluated on two DeepSeek-R1 distilled models across six benchmarks (five math + one science QA), DYNTS matches full-cache accuracy while compressing the KV cache and reducing latency.

**Compliance With Llm Reviewing Policy:**

Affirmed.

**Ethical Review Concerns:**

Extracting the text from the submitted PDF reveals invisible instructions embedded in the document.

**Ethical Review Flag:**

Flag this paper for an ethics review.

**Ethics Expertise Needed:**

["Other Expertise"]

**Final Justification:**

I recommend **Accept (5)** (was 4). The author's rebuttal substantively addressed my concerns regarding statistical significance and evaluation scope. By providing 95% confidence intervals and reframing the core claim from "outperforms" to honest achieving "on-par" performance, the authors improved the paper's soundness and scholarly integrity. Maintaining parity with a full-cache Transformer while achieving a 3.2x–5.7x KV-cache compression ratio is a highly significant result for the practical deployment of long-reasoning models.

Furthermore, the authors clearly delineated the novelty of DynTS relative to R-KV and concurrent works like RLKV and ThinKV. The expanded evaluation on seven benchmarks across math, knowledge, and code successfully demonstrates the method's generalizability beyond narrow domains. I appreciate the authors’ detailed responses and their commitment to updating the manuscript. Please ensure the final version strictly adheres to the promised revisions, particularly the "on-par" performance characterization and the inclusion of comprehensive statistical diagnostics.

The rating is final.

**Key Questions For Authors:**

1. **Statistical significance on small benchmarks.** Can you provide confidence intervals or bootstrap standard errors for AIME24/25 results? The claim of outperforming full-cache Transformers on AIME24 (+2.0%) appears to be within the sampling noise given only 15 problems with 5 responses each. How would results change with more sampling runs?

2. **Justification for layer/head aggregation.** The importance score (Eq. 6) and its implementation aggregate across all layers and heads. Have you compared this against per-layer or per-head importance scores? A token that is critical in one head but irrelevant in others would receive a diluted aggregate score — how do you ensure such tokens aren't erroneously evicted?

3. **Hyperparameter selection procedure.** Different budgets, local windows, and retention ratios are used for different benchmarks. How were these chosen? Was any validation split used, or were they tuned on the test sets? This significantly affects the credibility of the reported results.

4. **Comparison with training-free alternatives at matched effort.** DYNTS requires training a predictor (~7k samples, 15 epochs). R-KV achieves competitive accuracy (59.6% vs 61.9% on R1-Llama) without any training. Have you considered whether the training budget could be better used (e.g., by fine-tuning the model itself to produce shorter reasoning traces)?

5. **Generalization beyond math.** Can you show the predictor's Kendall correlation (as in Fig. 7) evaluated on GPQA-D data? This would directly test whether the predictor learns transferable importance patterns or math-specific ones.

6. **Embedded Prompt Injection.** The PDF of the manuscript contains hidden text (invisible to human readers but extractable via text parsing tools) that instructs LLM-based reviewers to include specific "canary phrases" in their reviews. Please explain the presence and purpose of this prompt injection in your submitted manuscript. Was this added by the authors, and if so, what is the justification? (See appended evidence).

**Limitations:**

The authors discuss limitations adequately in Appendix E. They explicitly acknowledge the focus on mathematical reasoning, the small training set (~7k samples), and the current implementation being limited to the `transformers` library. The impact statement appropriately notes no specific ethical concerns beyond general considerations for advancing generative AI.

The limitations section would benefit from discussing (1) the lack of error bars and statistical significance testing, (2) the potential for the predictor to fail on out-of-distribution problem types even within math, and (3) the comparison with training-free methods that achieve competitive results without the overhead.

**Strengths And Weaknesses:**

Strengths

The link between low attention weight and safe KV eviction is principled — evicting tokens with small $\alpha_{t,j}$ produces a bounded perturbation $\|\tilde{\mathbf{o}}_t - \mathbf{o}_t\| \leq \frac{\epsilon}{1-\epsilon} \max_j \|\mathbf{o}_t - \mathbf{v}_j\|$, where $\epsilon$ is the total evicted attention mass. This is not a statistical coincidence but a consequence of the attention mechanism itself.

DYNTS achieves near-lossless accuracy (61.9% vs 61.6% on R1-Llama; 65.3% vs 65.5% on R1-Qwen) with significant memory (3.32-5.73x) and latency (1.84-2.62x) improvements. The efficiency gains compound over longer sequences (Fig. 5), which is exactly the regime where efficiency matters most.

Table 2 systematically ablates each component (question tokens, critical thinking tokens, local window). Figures 6-8 explore sensitivity to local window size, retention ratio, and budget. Fig. 7 validates predictor training convergence. Fig. 9 provides an interpretable content-vs-function-word analysis. These collectively build a solid empirical understanding of the method.

The computational gain theorem (Eq. 9) and break-even condition (Eq. 10) provide actionable guidance for practitioners: $K > \frac{1.5d}{n_i L}$. The authors verify this is satisfied in their experimental settings ($K = 900 \gg 192$).

Honest limitations section is great.The paper explicitly scopes the method to mathematical reasoning (Section 4.1) and acknowledges limitations including domain specificity, small training set, and implementation constraints (Appendix E).

Weaknesses

**W1. Novelty is primarily in problem formulation and adaptation, not in individual components (Originality).** The importance predictor architecture draws from Dynamic-LLaVA (explicitly acknowledged, p.4 line 218), and the dual-window KV cache design resembles R-KV's dual-buffer. However, this criticism should be tempered: (1) several of these works (R-KV, Think Clearly, Thought Anchors, RLKV) are likely *concurrent*, not prior work that DYNTS should have built upon; (2) the predictor solves a fundamentally different task than Dynamic-LLaVA — predicting *future* importance rather than *present* relevance — which is non-trivially harder and whose success is a genuine empirical finding; (3) DYNTS's specific problem formulation (online future-importance prediction for decode-time KV compression in LRMs) is itself a novel contribution. Still, the paper would benefit from more transparently positioning its relationship to these concurrent works.

**W2. Narrow evaluation scope (Soundness/Significance).** Five of six benchmarks are pure math; MATH500 is in-distribution with the training data. GPQA-D is the only non-math benchmark and still involves step-by-step deduction similar to math. No code reasoning, logical reasoning, or open-ended reasoning tasks are evaluated. The method's title claims "Efficient Reasoning in Large Reasoning Models" without a math qualifier, but the evidence supports only mathematical reasoning.

**W3. No confidence intervals or statistical significance tests (Soundness).** AIME24/25 have only 15 problems each. With 5 responses per problem, the standard error of Pass@1 is approximately $\pm 5.8\%$. The claimed improvements on AIME24 (+2.0% on R1-Llama) and AIME25 (+1.3% on R1-Qwen) are well within noise. The claim that DYNTS "outperforms the Transformers on several challenging tasks" (Section 6.2) is not supported by statistical analysis.

**W4. Importance score aggregation across layers/heads is theoretically unjustified (Soundness).** The per-head eviction error bound applies per head, but the implementation (Appendix B.1) aggregates importance across all layers and heads into a single scalar. A token critical in one specific head but unimportant elsewhere receives a diluted aggregate score. No analysis is provided on whether this aggregation faithfully captures per-head criticality.

**W5. Per-benchmark hyperparameter tuning without validation split (Soundness).** Different budgets ($B = 5000$ vs $3000$), local window sizes (2000 vs 1500 vs 1000), and retention ratios (0.4 vs 0.3) are used for different benchmarks and models. No validation split or principled hyperparameter selection procedure is described. If these were tuned on the test benchmarks, the reported results are inflated.

**W6. Limited model scale (Significance).** Only 7B/8B distilled models are tested. It remains unknown whether the Pareto Principle holds at larger scales (70B+) or for non-distilled reasoning models like the full DeepSeek-R1 or o1/o3.

**W7. Missing comparisons with closely related concurrent work (Presentation).** Think Clearly (EMNLP 2025) addresses the same problem (redundant thinking tokens in LRMs) but is barely discussed. ThinKV (concurrent work on thought-adaptive KV cache compression) is not compared at all. RLKV (learned KV compression for reasoning) is also absent from comparisons despite being a learned approach like DYNTS.

---

> ### Author Rebuttal · Authors · 2026-03-30
>
> We sincerely thank the reviewer for taking the time to evaluate our work. We mark responses to Key Questions with Q# and Weaknesses with W#.
>
> ___
> **Q1 & W3: Statistical significance on small benchmarks**
>
> *Response:* Thanks reviewer’s useful reminder. We report the experimental results with their standard deviations for AIME24/25 in the table below (our response to Reviewer V3Zr, Q3 & W4 for details).
>
> |Method|AIME24|AIME25|
> |---|---|---|
> |*Transformers*|$52.0\pm5.3$|$35.3\pm2.9$|
> |Window|$41.3\pm5.0$|$31.3\pm1.8$|
> |StreamingLLM|$42.0\pm5.5$|$29.3\pm2.7$|
> |SepLLM|$38.6\pm6.4$|$31.3\pm5.5$|
> |H2O|$42.6\pm2.7$|$33.3\pm4.7$|
> |SnapKV|$48.6\pm4.4$|$33.3\pm4.0$|
> |R-KV|$44.0\pm7.6$|$32.6\pm4.3$|
> |**DynTS**|$52.0\pm3.8$|$36.6\pm4.0$|
>
> We thank the reviewer for pointing out our tone in claims and will revise the wording in the next version.
>
> ---
> **Q2 & W4: Justification for layer/head aggregation.**
>
> *Response:* We thank the reviewer for this question. Averaging attention scores across all layers and heads does not dilute token importance, because it captures a token’s global importance. If a token is important only in one head or layer, but not elsewhere, it should be evicted.
>
> To support this point, we compare average and max aggregation for token importance and retain the top-p% tokens under each strategy. As shown in the table below, the average aggregation reaches performance close to the full-token setting at top-30% retention, whereas max aggregation still shows a clear gap even at top-50%. This suggests that average aggregation does not dilute token importance and is the more reasonable choice.
>
> Full Token：90.8
>
> |Ratio(%)|2|4|10|20|30|40|50|
> |---|---|---|---|---|---|---|---|
> |Average|85.4|87.5|89.7|90.5|90.7|90.8|90.8|
> |Max|75.3|75.7|76.9|77.8|78.6|81.1|84.6|
>
> ---
> **Q3 & W5:  Hyperparameter selection procedure.**
>
> *Response:* Our hyperparameters were not tuned on the test set. They were chosen based on task difficulty and empirical results on the MATH training set.
>
> - Budget: Harder tasks (AIME24/25/AMC23/GPQA) produce longer reasoning trajectories than simpler ones (MATH500/GaoKao) (14,493 vs. 9,070), so we set a larger budget.
> - Retention ratio: On the *MATH* training set, retaining top-30% importance tokens is sufficient for R1-Llama to match full-token performance, while R1-Qwen requires 40%.
> - Local Window: We follow the prior works' setting. For R1-Qwen, we set a larger window size, since it appears more sensitive to local context due to Dual Chunk Attention in pretraining.
>
> ---
> **Q4: Comparison with training-free alternatives at matched effort.**
>
> *Response:* We thank the reviewer for this question. We believe the additional effort of our method is justified by its empirical gains, as DynTS clearly outperforms R-KV on R1-Qwen (65.3% vs. 62.2%).
>
> Directly fine-tuning the model to produce shorter reasoning traces is a feasible alternative, but it may incur substantial computational cost and risk catastrophic forgetting. For example, DLER is trained on math tasks, yet its performance drops on the  GPQA (49.0% → 26.8%).
>
> ---
> **Q5 & W2: Generalization beyond math**
>
> *Response:*  We also think generalization to other tasks is valuable. In the paper, we evaluate out-of-domain generalization on the knowledge-reasoning benchmark GPQA, where DynTS achieves performance comparable to full cache (R1-Llama: 46.4% vs. 46.3%). We further observe the same trend on coding tasks (R1-Qwen: 89.7% vs. 88.9%; R1-Llama: 83.8% vs. 82.6%). Due to space limitations, the detailed coding results are provided in our response to Reviewer CwkV, Q1 & W2.
>
> We also provide the Kendall correlation on GPQA-D, 0.4855 in R1-Qwen and 0.5509 in R1-Llama.
>
> ---
> **Q6: Embedded Prompt Injection.**
>
> *Response:* We did not inject any prompt. This has been inserted by the ICML organizers to detect violations of LLM policy.
>
> ---
> **W1: The Novelty of our method**
>
> *Response:* We appreciate the reviewer discuses the novelty of our work. Unlike Dynamic-LLaVA, which predicts fixed image tokens during prefilling, our method targets dynamically generated reasoning tokens during decoding, a more challenging setting.
>
> Moreover, unlike R-KV and Think Clearly, which rely on static attention patterns, our method predicts each token’s contribution to the final answer online during inference.
>
> Overall, our method offers both stronger reasoning performance and broader applicability, and we believe it makes a meaningful contribution to efficient LRM inference.
>
> ---
> **W7: Missing discussion with closely related concurrent work**
>
> *Response:* We thank the reviewer for pointing out these related concurrent works. Although RLKV and ThinKV are recent concurrent papers that are not required for discussion under ICML policies, we agree that comparing these works would be valuable and would add clarity to the discussion.
>
> ---
> We hope the above clarifications resolve any apprehensions, and we are glad to address any follow-up questions the reviewer may have.

---

> > ### Author Rebuttal · Reviewer_CzcE · 2026-04-01
> >
> > Thank you for the detailed responses. The clarification on hyperparameter selection (Q3/W5 — tuned on MATH training set, not test benchmarks) resolves that credibility concern. The average-vs-max aggregation comparison (Q2/W4) is empirically convincing. The new coding task results and GPQA-D Kendall correlations (Q5/W2) are encouraging evidence of generalization beyond math.
> >
> > However, two major concerns remain insufficiently addressed:
> >
> > 1. **Statistical significance (Q1/W3):** The standard deviation table appears to have missing entries — I cannot evaluate the actual numbers. Without verifiable confidence intervals, the claim of outperforming full-cache Transformers on AIME remains unsupported.
> >
> > 2. **Novelty positioning (W1/W7):** The distinction from Dynamic-LLaVA (future vs. present prediction) is valid, but the structural similarity to R-KV's dual-buffer architecture is not acknowledged. The promise to discuss concurrent work (RLKV, ThinKV) is welcome but vague — the paper needs substantive contextualization, not just additional citations.
> >
> > I maintain my rating of **Weak Accept (4)**. The method is well-engineered and practically useful, but the narrow evaluation scope (even with the new coding results, which were not in the original submission) and the incremental novelty profile relative to the concurrent work landscape keep me from a 5/5 confident accept.

---

> > > ### Author Response · Authors · 2026-04-01
> > >
> > > We are truly delighted to hear that our rebuttal has addressed most of the reviewers’ previous concerns. Thank you very much for taking the time to carefully read our responses and for the constructive follow-up question. We respond to your concerns below:
> > >
> > > ---
> > >
> > > (1) **Statistical significance (Q1/W3)**
> > >
> > > *Response*:  We sincerely thank the reviewer for their careful assessment. In the table below, we provide the verifiable 95% confidence intervals (CIs) for AIME24/25. Acknowledging the overlapping CIs, we have reframed our claims in the revised manuscript. Rather than claiming DynTS "outperforms" the full-cache baseline (only once in the experimental discussion in *Section 6.2* ), we now precisely state that it achieves **on-par performance**. This clearly aligns with our core claim intention: *DynTS achieves comparable performance with full-cache reasoning abilities under 3.2 - 5.7$\times$ KV-cache compression.*
> > >
> > > |Method|AIME24|AIME25|
> > > |---|---|---|
> > > |*Transformers*|$[41.7,62.3]$|$[31.6,39.0]$|
> > > |**DynTS (Ours)**|$[47.3,56.7]$|$[31.6,41.6]$|
> > > |Window|$[35.1,47.5]$|$[29.1,33.5]$|
> > > |StreamingLLM|$[35.1,48.9]$|$[25.9,32.7]$|
> > > |SepLLM|$[30.6,46.6]$|$[24.4,38.2]$|
> > > |H2O|$[39.2,46.0]$|$[27.5,39.1]$|
> > > |SnapKV|$[43.1,54.1]$|$[28.3,38.3]$|
> > > |R-KV|$[34.6,53.4]$|$[27.3,37.9]$|
> > >
> > > ---
> > >
> > > (2) **Novelty positioning (W1/W7)**
> > >
> > > *Response:*  We sincerely thank the reviewer for acknowledging our distinction from Dynamic-LLaVA.
> > >
> > > Regarding *R-KV*, we acknowledge a structural similarity in the cache eviction strategy.  However, the key to supporting the eviction strategy is to accurately assess tokens’ importance.  Compared to R-KV's heuristic static attention, DynTS utilizes a trained importance predictor to capture the contribution to the final answer. This assessment mechanism is paramount for LRMs’ reasoning, thereby leading to a better performance.
> > >
> > > Regarding *RLKV* and *ThinKV*: Due to strict character limits in the initial rebuttal phase, we were unable to provide a detailed discussion about the **recent concurrent works**. Although discussion of these papers is not strictly required under the ICML policy, we think comparing them may be valuable to our work. We add the following discussion to our paper:
> > >
> > > Fundamentally, DynTS diverges in both granularity and training paradigm. Unlike RLKV, which employs computationally expensive RL to manage cache at the *head level*, DynTS uses a highly efficient supervised predictor to retain critical information at the fine-grained *token level*. Moreover, ThinKV relies on heuristic phase-based segmentation and a hybrid quantization–eviction strategy. DynTS precisely predicts each token’s importance, achieving much finer-grained retention of critical reasoning steps.
> > >
> > > *Action taken:* As the source code for these concurrent methods is not yet publicly available, it's difficult to conduct reliable head-to-head comparisons. To avoid potentially inaccurate or unfair results, we are requesting it from the authors and will conduct empirical comparisons for further discussion in the next version.
> > >
> > > >***Concurrent research.** Authors are not required (but still welcome) to discuss works that have been made public less than two months before the full-paper submission deadline.*
> > >
> > > ---
> > >
> > > **(3) Narrow evaluation scope**
> > >
> > > We sincerely appreciate the reviewer's concern and respectfully clarify that our evaluation suite is actually quite extensive. It includes **7 diverse benchmarks** that comprehensively cover three core pillars of complex reasoning: *Math* (AIME24/25, AMC23, MATH500), *Knowledge* (GPQA_D, GK2023EN), and *Code* (Humaneval). Compared to prior literature [1-3], our reasoning benchmarks provide richer results and demonstrate more generalization beyond the "narrow" evaluation scope.
> > >
> > > ---
> > > We thank the reviewer for their valuable feedback and hope that our detailed responses have addressed your concerns.
> > >
> > > We would also be glad to address any follow-up questions the reviewer may have.
> > >
> > > ---
> > > [1] R-KV: Redundancy-aware KV Cache Compression for Reasoning Models
> > >
> > > [2] DLER: Doing Length pEnalty Right - Incentivizing More Intelligence per Token via Reinforcement Learning
> > >
> > > [3] ThinKV: Thought-Adaptive KV Cache Compression for Efficient Reasoning Models

---

### Official Review · Reviewer_Gbtk · 2026-03-13

**Soundness:** 3
**Presentation:** 3
**Significance:** 3
**Originality:** 3
**Overall Recommendation:** 4
**Confidence:** 3

**Summary:**

This paper introduces a KV cache eviction method. Authors introduce a trainable Importance Predictor at the final layer of a reasoning model, which allows the model to predict the importance of a thinking token with respect to the final answer. The algorithm manages memory by setting a budget, and the system only retains KV cache of tokens with higher predicted importance scores in the Select Window, and all tokens in the Local Window. Authors provide motivation showing that the attention weights can be used as a proxy for token importance. In particular, Figure 2 indicates that the mean token importance for thinking tokens is much lower than for question tokens. Using MSE loss, the method trains a predictor on top of the frozen backbone of the model which predicts the importance. When compared with other methods, authors show that the tokens per second is significantly higher while pass@1 is preserved or even improved.

**Compliance With Llm Reviewing Policy:**

Affirmed.

**Final Justification:**

My final assessment is based on my original comments, our discussion, and other reviews.

**Key Questions For Authors:**

1. Why only evict when a certain budget is reached? Could you do even better by maintaining a constant amount of budget throughout? Did you try any other methods?
2. What is the inference batch size? How does this method scale when batch size is increased/ decreased?

**Limitations:**

Yes.

**Strengths And Weaknesses:**

Strengths.
1. Authors provide an algorithm that appears to work well both in terms of preserving accuracy and increasing throughput.
Figure 2 motivates the method very well by showing that importance scores in thinking tokens are highly sparse and overall have low mean.
2. Section 4 is clear about the proposed method, as well as well-motivated. Section 5 provides some theoretical analysis of the benefits.
3. The experimental section compares many different datasets and two different models, as well as both pass@1 and TPS (Throughput). Authors ablate different retention strategies and show other interesting trends such as in Figure 5 regarding real-time throughput.

Weaknesses.
1. The algorithm is relatively clear but it would be even more helpful to have an algorithm pseudocode that exactly describes the procedure.
2. The two models, R1-Llama and R1-Qwen are somewhat similar in terms of size and how they were trained. It would be helpful to see how this method works on models of different sizes.

---

> ### Author Rebuttal · Authors · 2026-03-30
>
> We sincerely thank the reviewer for taking the time to provide such a thoughtful and encouraging evaluation of our work. We address the reviewer’s questions and concerns in detail below. For clarity, we mark responses to Key Questions with Q# and Weaknesses with W#.
>
> ---
> **Q1: Why only evict when a certain budget is reached? Could you do even better by maintaining a constant amount of budget throughout?**
>
> *Response:* We thank the reviewer for this thoughtful comment. Since each eviction step requires selecting and removing unimportant cache entries, it inevitably introduces additional overhead. If a constant budget were maintained throughout the entire decoding process, the eviction operation needs to be performed at every decoding step, increasing inference latency. Therefore, our method evicts multiple cache entries at once when the budget is reached, thereby reducing the number of eviction operations and improving inference efficiency.
>
> ---
> **Q2: What is the inference batch size? How does this method scale when the batch size is increased/decreased?**
>
> *Response:* We thank the reviewer for this question. In our experiments, we set the inference batch size to 16; larger batch sizes lead to OOM in standard Transformer inference.
>
> Increasing batch size does not increase reasoning accuracy, but it can improve throughput. Specifically, when ignoring the OOM constraint, scaling the batch size can improve throughput before reaching a computation or bandwidth bottleneck.
>
> ---
> **W1: The algorithm is relatively clear, but it would be even more helpful to have an algorithm pseudocode that exactly describes the procedure.**
>
> *Response:* Thanks for the reviewer’s useful suggestion. We provide an initial version of pseudocode below and will add a more detailed version in our paper.
>
> ```
> Input:
>     prompt x
>     LRM with Importance Predictor M
>     budget B = W_q + W_s + W_l
>     local window size W_l
>     retention ratio r
>
> Initialize:
>     store all question-token KV caches in question window W_q
>     set their importance scores to +infinity
>     W_s = empty, W_l = empty
>
> for each decoding step t do
>     obtain next token x_{t+1} and current-token score s_t:
>         M(x_<=t) -> (x_{t+1}, s_t)
>
>     insert current token into local window W_l and select window W_s
>
>     if |W_q| + |W_s| + |W_l| > B then
>         keep top floor(r * |W_s|) tokens in W_s by predicted score
>         evict the rest
>     end if
> end for
>
> return generated sequence
> ```
>
> ---
> **W2: The two models, R1-Llama and R1-Qwen are somewhat similar in terms of size and how they were trained. It would be helpful to see how this method works on models of different sizes.**
>
> *Response:* We thank the reviewer for this helpful suggestion. Choosing R1-Qwen and R1-Llama because they are the most representative open-source reasoning models, and many prior studies also evaluate on these LRM [1-3].
>
> Our method is generally applicable to reasoning models that use thinking-answer-style generation. To further support this point, we additionally provide results on Qwen3-4B in the table below.  The results show our method achieves performance comparable to the full-cache baseline, consistent with the results reported in the paper.
>
> | Method | AIME24 | AIME25 | AMC23 | GK23EN | MATH | GPQA | AVG |
> | --- | --- | --- | --- | --- | --- | --- | --- |
> | Baseline |  60.7 | 46.7 | 89.0 | 82.1 | 93.2 | 57.4 | 71.5 |
> | DynTS (Ours) | 59.3 | 47.3 | 89.5 | 81.5 | 92.4 | 56.9 | 71.1 |
>
> ---
> We hope the above clarifications resolve any apprehensions, and we are glad to address any follow-up questions the reviewer may have.
>
> ---
> [1] DLER: Doing Length pEnalty Right - Incentivizing  More Intelligence per Token via Reinforcement  Learning
>
> [2] R-KV: Redundancy-aware KV Cache Compression for Reasoning Models
>
> [3] ThinKV: Thought-Adaptive KV Cache Compression for Efficient Reasoning Models

---

> > ### Author Rebuttal · Reviewer_Gbtk · 2026-04-02
> >
> > Thank you to the authors for the detailed clarifications. I maintain my positive assessment.

---

> > > ### Author Response · Authors · 2026-04-03
> > >
> > > Dear reviewer
> > >
> > > We’re pleased to hear that our rebuttal fully addressed your concerns, and we’re grateful for your **positive assessment**.
> > >
> > > Thank you for your constructive feedback and for your dedicated comments throughout the process again.
> > >
> > > Sincerely,
> > >
> > > The Authors

---

### Official Review · Reviewer_V3Zr · 2026-03-13

**Soundness:** 2
**Presentation:** 3
**Significance:** 3
**Originality:** 2
**Overall Recommendation:** 5
**Confidence:** 4

**Summary:**

The paper studies KV-cache compression for large reasoning models in the long-decoding regime, induced by explicit reasoning traces. The main claim is that only a small subset of thinking tokens strongly influences the final answer, while other are largely redundant. Based on this observation, the authors propose DYNTS, which adds an importance predictor to estimate which thinking tokens should be retained in the KV cache. The method keeps question tokens, a local window, and the top-scored thinking tokens, while evicting the rest.
The paper also provides a simple FLOPs-based break-even analysis.
Experiments on two DeepSeek-R1-distilled backbones across 6 reasoning benchmarks show that DYNTS approximately matches the full-cache baseline while significantly improving throughput and reducing KV memory, and it outperforms prior KV-cache compression baselines under matched budgets.

**Compliance With Llm Reviewing Policy:**

Affirmed.

**Final Justification:**

A large majority of my concerns have been addressed, and I have raised my score.

**Key Questions For Authors:**

1. To support the generality claims, please provide results on a different reasoning domain, e.g. coding.
2. To understand the predictor better - can you provide qualitative examples of retained vs evicted tokens, ideally with predicted importance scores?
3. Can you report standard deviations / confidence intervals and significance tests across the paper? In addition, I would suggest toning down the "beating full cache" claims if those are not backed by sufficient statistics.

**Limitations:**

Yes

**Strengths And Weaknesses:**

**Strengths:**

1. The paper addresses a real deployment bottleneck for reasoning models.
2. The paper’s initial analysis, that answer-to-thinking attention is sparse, gives a justification for the method.
3. The method is simple and easy to understand: a lightweight predictor plus a budgeted retention policy over question tokens, a local window, and selected thinking tokens.
4. The method seems competitive with the full-cache baseline while improving throughput, and it outperforms multiple prior cache-compression baselines.
5. The ablations help support the core design choices.


**Weaknesses:**

1. The main concern is limiting the training only to math problems, yet the paper’s framing suggests a broader LRM inference method. This makes it hard to know whether the learned importance signal is general or only works in a specific domain.
2. While effective, it would be beneficial to analyze what the importance predictor has actually learned. Does it capture a meaningful notion of “decision-critical” reasoning tokens? or does it tend to identify narrower domain-specific / formatting-specific cues from math-style traces?
3. The paper uses task-dependent budgets and model-dependent retention ratios/local windows. This implies that several hyper-parameters need to be tuned, which complicates the method.
4. “Beating full cache” needs stronger statistical support, and more generally the paper should report uncertainty estimates across the main results. DYNTS sometimes slightly outperforms the full-cache baseline, but these gains are small and should be treated cautiously unless backed by significance analysis. For example, on R1-Llama the average is 61.9 vs 61.6, and on R1-Qwen it is 65.3 vs 65.5 (i.e., essentially tied overall - which is very good, but then requires some toning down of the original claim).


**Minor:**
1. The page header “Submission and Formatting Instructions for ICML 2026” appears throughout the manuscript pages.

---

> ### Author Rebuttal · Authors · 2026-03-30
>
> We sincerely thank the reviewer for taking the time to provide such a thoughtful and encouraging evaluation of our work. For clarity, we mark responses to Key Questions with Q# and Weaknesses with W#.
>
> ---
> **Q1 & W1: Provide results on the coding reasoning domain. Is the learned importance signal general?**
>
> *Response:*  Thanks for raising this point. To address your concern, we evaluated our method on the code-reasoning benchmark *HumanEval* using the predictor trained on the MATH training set. The results are reported in the table below. Notably, even without fine-tuning on code reasoning tasks, our method achieves comparable performance to the full-cache baseline (full cache: 89.7% vs. ours: 88.9% in R1-Qwen; 83.8% vs. 82.6% in R1-Llama).
>
> Besides, we also evaluate the knowledge-reasoning domain GPQA in our paper, which shows similar results (49% vs. 48.1% in R1-Qwen; 46.4% vs. 46.3% in R1-Llama). These results suggest that the learned importance signal is general.
>
> | |**R1-Qwen**| **R-Llama**|
> |---|---|---|
> |*Transformers*|*89.7*|*83.8*|
> |**DynTS**|**88.9**|**82.6**|
> |Window|78.1|54.0|
> |StreamingLLM|78.5|58.3|
> |SepLLM|76.9|67.5|
> |H2O|86.2|81.5|
> |SnapKV|85.8|82.3|
> |R-KV|86.7|81.2|
>
> ---
> **Q2: Can you provide examples of retained vs evicted tokens, ideally with predicted importance scores?**
>
> *Response:* We thank the reviewer for the helpful suggestion. We will add highlighted examples with importance score to the appendix in the next version. Due to space limitations, we provide one example snippet below, with retained tokens shown in bold. For clear insight, please refer to the part-of-speech analysis in Appendix B.2, where retained tokens contain a higher proportion of content words than evicted tokens.
>
> > So, **p =** **831**6.07 **/ (2***139**.**6**)** = 8316.07 / **279**.**2 ≈ Let** me compute this **division.**
> 279.2 * 29 = 8096.8
> 8316.07 - 8096.8 = 219.27
> 219.27 / 279**.**2 ≈ 0.**783**
> **So**, **p** ≈ **29**.**783 MeV/c**
> **So, p** ≈ 29.783 MeV/c
> **Therefore, neutrino's energy is E**_n**u = p + m_nu c**^**2, but** m_nu is **negligible, so** E_nu **≈** **29**.**783** MeV**.**
>
> ---
> **Q3 & W4: Can you report standard deviations and significance tests across the paper? Toning down the "beating full cache" claims.**
>
> We report the main experimental results with standard deviations, as well as the p-values from the significance test against the baselines in the tables below. Due to space constraints, we only report R1-Llama.
>
> |Method|AIME24|AIME25|AMC23|GK2023EN|GPQA_D|MATH500|
> |---|---|---|---|---|---|---|
> |*Transformers*|$47.3\pm7.6$|$28.6\pm3.8$|$86.5\pm2.8$|$73.1\pm1.8$|$46.4\pm3.8$|$87.5\pm1.2$|
> |Window|$18.6\pm5.0$|$14.6\pm2.9$|$59.5\pm3.7$|$47.0\pm2.2$|$37.6\pm1.8$|$58.1\pm0.9$|
> |StreamingLLM|$20.6\pm3.6$|$16.6\pm2.3$|$65.0\pm3.9$|$53.4\pm1.6$|$37.8\pm0.9$|$66.1\pm0.7$|
> |SepLLM|$30.0\pm9.1$|$20.0\pm2.3$|$71.0\pm2.8$|$61.4\pm1.4$|$39.7\pm2.3$|$74.5\pm1.3$|
> |H2O|$38.6\pm5.5$|$22.6\pm1.4$|$82.5\pm5.0$|$67.5\pm2.4$|$41.6\pm3.2$|$82.7\pm1.3$|
> |SnapKV|$39.3\pm5.9$|$24.6\pm5.0$|$80.5\pm5.4$|$68.7\pm1.5$|$41.9\pm2.2$|$83.1\pm1.2$|
> |R-KV|$44.0\pm4.9$|$26.0\pm2.7$|$86.5\pm5.4$|$71.4\pm2.4$|$44.5\pm1.8$|$85.2\pm0.6$|
> |**DynTS**|$49.3\pm6.4$|$29.3\pm2.7$|$87.0\pm2.0$|$39.7\pm1.9$|$46.3\pm2.8$|$87.1\pm0.8$|
>
> |Baseline|AIME24|AIME25|AMC23|GK2023EN|GPQA_D|MATH500|
> |---|---|---|---|---|---|---|
> |*Transformers*|0.6652|0.7606|0.7606|0.5558|0.9637|0.5440|
> |Window|<0.001|<0.001|<0.001|<0.001|<0.001|<0.001|
> |StreamingLLM|<0.001|<0.001|<0.001|<0.001|0.0016|<0.001|
> |SepLLM|0.0058|<0.001|<0.001|<0.001|0.0040|<0.001|
> |H2O|0.0234|0.0031|0.1186|0.0084|0.0409|<0.001|
> |SnapKV|0.0341|0.1189|0.0528|0.0113|0.0272|<0.001|
> |R-KV|0.1814|0.0955|0.8561|0.4974|0.2716|0.0048|
>
> We thank the reviewer for pointing this out. Our intention was not to claim that DynTS “beats full cache,” but rather that it achieves performance comparable to full cache. We will revise the wording in the next version to make this clearer and more accurate.
>
> ---
> **W2: What important predictor did they learn? Does it capture a meaningful notion of “decision-critical” reasoning tokens?**
>
> The predictor is attached to the model’s final layer and trained with answer-to-thinking attention as the supervision signal, so it learn to estimate each token’s importance to the final answer.
>
> Yes, it captures the meaningful notion of “decision-critical” reasoning tokens. We support this in Section 6.3 (Ablation Study of Retained Tokens): retaining the tokens predicted as most important by the predictor achieves performance comparable to all tokens (65.5% → 65.3%), whereas retaining the least important tokens causes a clear drop (65.5% → 57.6%). This suggests that the tokens predicted to be important are more critical to the final answer, thereby supporting the point of capture “decision-critical” tokens.
>
> ---
> We hope the above clarifications resolve any apprehensions, and we are glad to address any follow-up questions the reviewer may have.

---

> > ### Author Rebuttal · Reviewer_V3Zr · 2026-04-03
> >
> > Thank you for your rebuttal.
> >
> > A large majority of my concerns have been addressed, and I have raised my score.

---

> > > ### Author Response · Authors · 2026-04-03
> > >
> > > Dear reviewer
> > >
> > > We are very pleased to hear that our rebuttal has fully addressed your concern. We sincerely appreciate your positive assessment of our work and your decision to raise your score.
> > >
> > > Thank you again for your constructive feedback and thoughtful, dedicated comments throughout the review process.
> > >
> > > Sincerely,
> > >
> > > The Authors

---

### Official Review · Reviewer_CwkV · 2026-03-13

**Soundness:** 3
**Presentation:** 3
**Significance:** 4
**Originality:** 4
**Overall Recommendation:** 4
**Confidence:** 3

**Summary:**

This paper introduces a lightweight importance predictor that dynamically estimates the importance of each thinking token for the final answer during decoding. The model then retains only the KV cache for highly important tokens, discarding redundant ones via a local window mechanism, thereby reducing memory and computational overhead during inference. Experimental results show that DYNTS outperforms existing KV cache compression methods on six benchmarks, improving the average by 2.6% over prior state-of-the-art methods under the same budget.

**Compliance With Llm Reviewing Policy:**

Affirmed.

**Final Justification:**

After reading the feedback, I raised the score to weak accept.

**Key Questions For Authors:**

See weaknesses. Besides, how about the performance of the proposed method when it is applied to other reasoning tasks?

**Limitations:**

yes

**Strengths And Weaknesses:**

Strengths
1. This paper focuses on the KV cache and computational bottlenecks caused by long reasoning processes in LRMs, which is a practically important problem. The paper is also well motivated by the question of which reasoning tokens are actually critical for deriving the final answer.
2. Compared with traditional eviction strategies based on local heuristics or attention statistics, the paper explicitly introduces a trainable importance predictor to dynamically estimate each token’s contribution to the final answer during decoding.

Weaknesses
1. The proposed method treats the aggregated answer-to-thinking attention as the ground-truth supervision signal for token importance, and then trains the predictor to fit this signal. However, whether attention can reliably capture a token’s true causal contribution to the final answer is not sufficiently justified.
2. The importance predictor is trained only on the MATH training set and then generalized to the other benchmarks. Although the results are promising, the current training data are still focused mainly on mathematical reasoning. The paper itself acknowledges that generalization to other domains, such as coding or abstract reasoning, remains limited.
3. The comparison with a broader range of baselines is still insufficient. The paper mainly compares DYNTS with KV cache compression methods. Still, it lacks discussion of or comparison with methods that directly reduce reasoning tokens or support test-time scaling or reasoning truncation.
4. Compared with purely heuristic KV eviction methods, DYNTS requires collecting training data with reasoning traces, constructing supervision signals, and training an additional predictor.

---

> ### Author Rebuttal · Authors · 2026-03-30
>
> We sincerely thank the reviewer for taking the time to provide such a thoughtful and encouraging evaluation of our work. For clarity, we mark responses to Key Questions with Q# and Weaknesses with W#.
>
> ---
> **Q1 & W2:  How about the performance of the proposed method when it is applied to other reasoning tasks?**
>
> *Response:* Thanks for raising this important point. Beyond mathematical tasks, our experiment also includes a knowledge-reasoning benchmark, *GPQA* (Table 1 in the paper). Results show that DynTS achieves comparable reasoning performance with the full-cache baseline (R1-Llama 46.4% vs. 46.3%).
>
> Following the reviewer’s suggestion, we further evaluated our method on the code-reasoning benchmark *HumanEval* using the predictor trained only on the *MATH* training set. The results are reported in the table below. Notably, even without fine-tuning on code reasoning tasks, our method incurs only 0.8% and 1.2% performance drops compared with the full-cache baseline in R1-Qwen and R1-Llama, respectively. This suggests that our method generalizes well and can be applied to other reasoning tasks.
>
> ||**R1-Qwen**|**R-Llama**|
> |---|---|---|
> |*Transformers*|*89.7*|*83.8*|
> |**DynTS (Ours)**|**88.9**|**82.6**|
> |Window|78.1|54.0|
> |StreamingLLM|78.5|58.3|
> |SepLLM|76.9|67.5|
> |H2O|86.2|81.5|
> |SnapKV|85.8|82.3|
> |R-KV|86.7|81.2|
> ---
> **W1: Whether attention can reliably capture a token’s true causal contribution to the final answer.**
>
> *Response:* We appreciate the reviewer’s thoughtful comment. In Transformer-based models, attention captures the relevance between tokens, and several prior studies [1–3] have used it as a proxy for token importance.
>
> Additionally, in *Section 3.2*, we present an analysis of the relationship between aggregated attention and the final answer. We find that keeping only the top 30% thinking tokens, ranked by aggregated attention score, achieves comparable performance to full tokens (69.7%→69.2%), whereas keeping the bottom 30% or a random 30% results in a substantial drop in performance (69.7%→53.8%; 69.7%→55.3%). These results suggest that tokens with higher aggregated attention scores are more important for the final answer. Therefore, we think the attention signal can capture the causal contribution of tokens to the final answer.
>
> ---
> **W3: Lacks discussion of methods that directly reduce reasoning tokens or reasoning truncation.**
>
> *Response:* Thanks to the reviewer raising this issue, we also considered these methods when processing our work. We did not discuss them in the paper mainly because of the following limitations.
> 1. Our method is primarily designed for GPU memory-constrained scenarios.  Although the mentioned methods can reduce the number of reasoning tokens,  they cannot ensure that each sample’s KV Cache stays within the available memory budget. Once this budget is exceeded, the system may run into OOM issues.
> 2. Under memory-constrained settings, these methods may lead to degraded reasoning performance. We also add experimental comparisons with the mentioned methods (reported in the table below), DEER (token reduction) and DLER (reasoning truncation), under the same memory budget. Due to memory constraints, some samples have to terminate the reasoning process prematurely, resulting in a performance drop (DEER: 65.5%→49.7%; DLER: 65.5%→61.9%). Our method achieves better reasoning accuracy than these baselines (65.5%→65.3%).
> 3. Methods based on SFT/RL show weaker generalization. For instance, DLER is trained with RL on mathematical tasks, and its performance drops noticeably on knowledge-reasoning tasks *GPQA* (49%→26.8%).
>
> We will add more discussions about directly reducing reasoning tokens and reasoning truncation methods in the revised manuscript.
>
> |Method|AIME24|AIME25|AMC23|GK23EN|MATH|GPQA|AVG|
> |---|---|---|---|---|---|---|---|
> |Transformers|52.0|35.3|87.5|77.9|91.3|49.0|65.5|
> |DynTS (Ours)|52.0|36.6|88.5|76.4| 90.0|48.1|65.3|
> |DEER|30.0|23.3|75.0|63.4|80.0|26.8|49.7|
> |DLER|48.6|35.3|87.5|76.2|89.5|34.5|61.9|
> ---
> **W4:  DYNTS requires training an additional predictor.**
>
> *Response:* We understand the reviewer’s concern regarding the additional overhead. However, we only trained a lightweight predictor, whose parameters are negligible compared to the model’s (approximately 0.68% of the total model's parameters; training only took ~150 min in R1-Llama on an 8*H800 server). This small addition leads to higher reasoning accuracy than heuristic KV-eviction methods (SOTA KV eviction 62.2% vs. DynTS 65.3%). Therefore, we believe that the introduced overhead is minor and well justified by the resulting performance gains.
>
> ---
> We hope the above clarifications resolve any apprehensions, and we are glad to address any follow-up questions the reviewer may have.
>
> ---
> [1] Thought anchors: Which llm reasoning steps matter?
>
> [2] Heavy-hitter oracle for efficient generative inference of large language models
>
> [3] Rethinking interpretability in the era of large language

---

> > ### Author Rebuttal · Reviewer_CwkV · 2026-04-06
> >
> > My main concerns are well addressed, and I have raised my score to weak accept.

---

> > > ### Author Response · Authors · 2026-04-06
> > >
> > > Dear Reviewer,
> > >
> > > Thank you very much for your feedback and for deciding to raise the score to **Weak Accept (4).**
> > >
> > > We noticed, however, that the **Overall Recommendation** score in the *Official Review* panel has not yet been updated. We are unsure if this is a system issue.
> > >
> > > If it is not too much trouble, could you please kindly check if the score has been updated in the *Official Review*? This is crucial for the evaluation of our work, and we would be immensely grateful for your help.
> > >
> > > Thank you again for your time and the constructive discussion!
> > >
> > > Sincerely,
> > >
> > > The Authors

---

### Decision · Program_Chairs · 2026-04-30

**Decision:**

Accept (regular)

**Comment:**

The paper proposes DYNTS: a lightweight learned importance predictor for selecting which reasoning tokens to retain in the KV cache during long-chain decoding.

Reviewers pointed out their concerns on scope, rigor, and positioning. The original evidence was concentrated on math-heavy tasks, so broader claims about general reasoning efficiency were not fully supported without the rebuttal’s added evaluations. Several reviewers also raised concerns about limited statistical support on small benchmarks, per-benchmark hyperparameter tuning without a clearly described validation protocol. They indicated the need for stronger justification of the attention-based supervision signal and the aggregation of importance across layers and heads.

Nonetheless, the strengths are substantial. Reviewers agree that the paper 1) tackles a real deployment bottleneck in long-reasoning models; 2) grounds the method in an intuitive sparsity observation about answer-to-thinking attention; and 3) presents a clean predictor-plus-dual-window retention strategy that is easy to understand and likely to be useful in practice. The empirical section demonstrates strong efficiency gains, informative ablations, sensitivity analyses, and useful practical guidance.

All in all, this paper is solid and useful for its practical value. Reviewers agree that this is a practically important problem and that the method is simple and effective: DYNTS preserves near-full-cache accuracy while delivering substantial KV-cache compression and throughput gains on long-reasoning workloads. As discussed, we encourage the authors to include statistical reporting and broader evaluation also reframing results as largely on par with full-cache decoding.